# A Study on PAVE Specification for Learnware

**Hao-Yu Shi, Zhi-Hao Tan, Zi-Chen Zhao, Yang Yu, Zhi-Hua Zhou**
National Key Laboratory for Novel Software Technology, Nanjing University, China
School of Artificial Intelligence, Nanjing University, China
`{shihy,tanzh,zhaozc,yuy,zhouzh}@lamda.nju.edu.cn`

## Abstract

"*Learnware = Model + Specification*". A learnware comprises a submitted model paired with a *specification* sketching its capabilities. For a Learnware Dock System (LDS) which accommodates numerous models, these specifications are essential to enabling users to identify helpful models, eliminating the requirement for prohibitively costly per-model evaluations. Recently, **Pa**rameter **Ve**ctor (PAVE) specification, which utilizes the changes in pre-trained model parameters to inherently encode the model capability and task requirements, shows promising capabilities in enabling identifying useful learnwares for high-dimensional, unstructured text data. In this paper, we present a comprehensive study of PAVE specification for learnware identification. Theoretically, from the neural tangent kernel perspective, we establish a tight connection between PAVE and prior specifications, providing a theoretical explanation for their shared underlying principles. We further approximate PAVE in a low-rank space and analyze the approximation error bound, highly reducing the computational and storage overhead. Extensive empirical studies demonstrate that PAVE specification excels at identifying CV and NLP learnwares even from heterogeneous learnware repository with corrupted model quality. Reusing identified learnware to solve user tasks can even outperform user-fine-tuned pre-trained models in data-limited scenarios.

## 1 Introduction

"*Learnware = Model + Specification*" (Zhou, 2016; Zhou and Tan, 2024). Consider a scenario where many machine learning developers are willing to share their trained models but are reluctant to disclose their private training data. In this context, we can envision a **Learnware Dock System (LDS)** (Tan et al., 2024b) that accommodates a diverse array of models. These models may be trained on different data, for different tasks, and based on different learning algorithms, w.r.t. different objectives, appearing as different models (not limited to neural networks), contributed by developers all over the world. Then, with the LDS accommodating a huge amount of models, a future user, who plans to build her own machine learning model, can submit her task requirement to the LDS, and LDS will then identify some helpful model or even reassemble a sets of models to be helpful to the user, such that she does not need to build her own model from scratch. Note that, during the whole process, neither model developers nor the user disclose their own training data to the LDS. Given the fact that the LDS does not have access to the original training data, and that many models are blackbox ones whose functionality can hardly be understand, how can the LDS identifies helpful models to the user task? how can the various models, even seemingly-irrelevant ones, be identified and reassembled to work together for the user?

"Learnware" makes this possible. A key ingredient is the "specification" which enables a trained model to be adequately identified to reuse according to the requirement of new user who knows nothing about the model, while model developers' training data are preserved. Briefly speaking, when each model is submitted to LDS, a specification generated by a machine learning process will be assigned to the model, such that the model becomes a learnware. The specification is able to roughly characterize the functionality of the model to some extent, without getting access to its training data. Future user's task requirement will also be submitted in the form of task specification. Once the LDS has accommodated a huge amount of learnwares, it is expected that many fancy abilities can

---

*Correspondence: Zhi-Hao Tan <tanzh@lamda.nju.edu.cn>

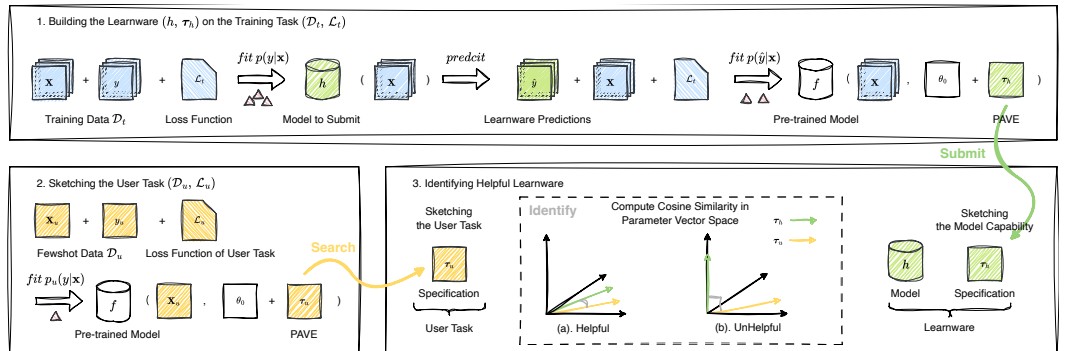

Figure 1: **Identifying helpful learnwares based on parameter vector similarity.** 1. The developer trains the model with a large amount of data in $\mathcal{D}_t$ and generates the model vector $\boldsymbol{\tau}_h$ based on the model prediction $\hat{y}$, then submits them to the system as a learnware. 2. The user generates the task vector $\boldsymbol{\tau}_u$ from a few samples in $\mathcal{D}_u$. 3. The larger the cosine similarity between the model and task vectors means that the more likely the model capability is to fulfill the user task requirements.

emerge, such as "small models do big" (Zhou and Tan, 2024; Tan et al., 2025), and models can exhibit abilities of doing things even beyond their original development purposes (Liu et al., 2024). Existing studies (Tan et al., 2024a; Lei et al., 2024) have shown success in tabular scenarios by using a provided explicit kernel to generate a privacy-preserving reduced set serving as a specification to sketch the data distribution where models excel. But when facing high-dimensional, unstructured data like images or text, the required sample complexity based on classic universal kernels tends to be unaffordable (Phillips and Tai, 2020).

Recently, by fine-tuning a shared pre-trained model to fit $p(\hat{y}|\mathbf{x})$ with the task's customized loss function, the changes in the pre-trained model's parameters, referred to as *Parameter Vector* (Tan et al., 2025), provide an effective way to represent the model capabilities and user task requirements. Fundamentally, a trained model can be viewed as a mapping from input $x$ to output $\hat{y}$, which is represented by $p(\hat{y}|\mathbf{x})$, thus the most straightforward way to sketch the conditional probability $p(\hat{y}|\mathbf{x})$ is to fit a model to it directly. During the process, information about $p(\hat{y}|\mathbf{x})$ essentially *leads—and only leads—to* changes in the pre-trained model's parameters, and is thus incorporated into the changes, which can characterize models and tasks in a unified way. Changes in model parameters have also been explored in Model Merging (Ilharco et al., 2023; Ortiz-Jiménez et al., 2023) and Gradient Matching (Zhao et al., 2021; Shi et al., 2022).

We find that the parameter vector also addresses some key difficulties for learnware identification in open and complex real-world scenarios. Specifically, in such scenarios, it is very challenging to identify helpful learnwares for user task due to numerous diverse **task types and semantics** and a lack of assurance regarding **model quality**. **(a)**. Due to the diversity of tasks, the output $y$ usually has different meanings and is difficult to encode universally, making their comparison intractable. For instance, models trained on a single face dataset can address diverse tasks, such as age prediction and emotion classification. However, it's hard to tell which model is best suited for hair color recognition. **(b)**. Regarding model quality, some low-quality models that might not be adequately trained on their respective training datasets yet are still submitted. Evaluating model quality using unified test datasets is unrealistic in open, real-world scenarios due to task diversity and data privacy concerns. For the above issues, the parameter vector shows a promising solution by sketching the model capability $p(\hat{y}|\mathbf{x})$ and capture both the task semantics and model quality in a unified way. Though, directly utilizing the parameter vector will result in unaffordable computational and storage cost. This motivates us to approximate the parameter vector in a low-rank space, enabling efficient identification of helpful learnwares from a large number of candidates for user tasks.

In this paper, we present a comprehensive study of **Pa**rameter **Ve**ctor (PAVE) specification for identifying learnwares proficient in user tasks by measuring the vectors similarity, shown specifically in Figure 1, showing its potential to quantify cross-task model-task compatibility—especially in

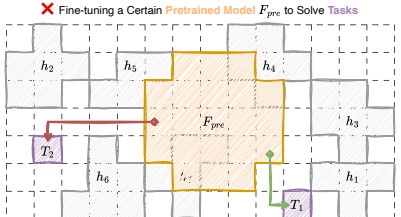 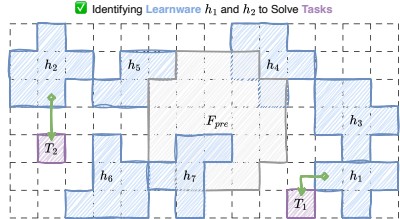 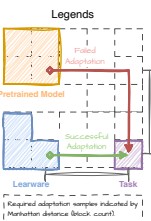

Figure 2: **Advantages over fine-tuning pre-trained models.** With limited samples and the restricted capacity of a single model, even a well pre-trained model $F_{pre}$ may fail to adapt to an unusual user task $T_2$. In contrast, a large collection of learnwares offers stronger overall capability, allowing more user tasks to be addressed by identifying appropriate models for further adaptation. Arrow length (Manhattan distance in blocks) denotes the required number of samples when adaptation is needed.

scenarios with heterogeneous tasks, involving diverse semantics and varying output spaces, or even across different model qualities. Our main contributions can be summarized as follows:

- We first formalize PAVE specification which constructs a compact representation based on the changes in pre-trained model parameters, inherently encoding the model capability and task requirements. Then we theoretically establish a tight connection between PAVE and RKME specification (Zhou and Tan, 2024) with a specific kernel, providing a theoretical explanation for their shared underlying principles.

- We further approximate the parameter vector in a low-rank space and analyze the approximation error bound, substantially reducing computational and storage overhead.

- Extensive experiments spanning CV and NLP reveal that the PAVE specification can effectively identify quality learnwares from heterogeneous and corrupted repositories, and these identified learnwares can even outperform user-fine-tuned models in data-limited scenarios.

## 2 PROBLEM SETUP

In this section, we summarize the learnware identification process for solving machine learning tasks. In general, we assume that each machine learning task $(\mathcal{D}, \mathcal{L})$ consists of a dataset $\mathcal{D} \subseteq \mathcal{X} \times \mathcal{Y}$ and a loss function $\mathcal{L} : \mathcal{Y} \times \mathcal{Y} \to \mathbb{R}$, where $\mathcal{X}$ is the input space and $\mathcal{Y}$ is the output space. And a learnware $(h, \boldsymbol{\tau}_h)$ consists of a model $h : \mathcal{X} \to \mathcal{Y}$ and a specification $\boldsymbol{\tau}_h$ for sketching the model capability. A learnware is helpful for the user task $(\mathcal{D}_u, \mathcal{L}_u)$ if the corresponding model achieves a low loss $\mathcal{L}_u$ on dataset $\mathcal{D}_u$, thereby performing well on the data distribution $p_u(\mathbf{x}, y)$ from which $\mathcal{D}_u$ is drawn.

The entire identification process of learnwares can be divided into three parts, shown in Figure 1:

1. For the $i$-th developer, they train the model $h_i$ on their local training dataset $\mathcal{D}_{t_i}$ using the loss function $\mathcal{L}_{t_i}$ and construct the corresponding specification $\boldsymbol{\tau}_{h_i}$. The learnware $(h_i, \boldsymbol{\tau}_{h_i})$ is then submitted to the learnware dock system, a platform for sharing learnwares.

2. For a user aiming to solve the task $(\mathcal{D}_u, \mathcal{L}_u)$, they construct a specification $\boldsymbol{\tau}_u$ based on the few-shot dataset $\mathcal{D}_u$ and the task-specific loss function $\mathcal{L}_u$. Instead of training a model from scratch, the user submits $\boldsymbol{\tau}_u$ to the platform to find suitable learnwares for their task.

3. The learnware dock system identifies learnwares that best match the user's task and minimizes the loss $\mathcal{L}_u$ on $p_u$ based on the specifications submitted by both users and developers.

## 3 METHODOLOGY

In this section, we formalize the PAVE specification for identifying helpful learnwares via parameter vector similarity. Section 3.1 presents the formalization of sketching the model capability by fitting the distribution $p(\hat{y}|\mathbf{x})$ for learnware identification. Section 3.2 then establishes the theoretical connection between PAVE and prior specifications (Zhou and Tan, 2024; Wu et al., 2023), showing

the underlying shared principles. Finally, Section 3.3 shows how parameter vectors can be optimized by compressing them into a low-rank representation, substantially reducing storage and computational overhead, and analyzes the approximation error bound theoretically.

## 3.1 FORMALIZATION OF PARAMETER VECTOR SPECIFICATION

To sketch model capability to identify high-quality learnware that closely aligns with user tasks, the parameter vector, serving as the specification, is constructed by fitting the distribution $p(\hat{y}|\mathbf{x})$, integrating both task semantics and model quality.

To underscore the importance of sketching a model's capabilities, we begin with an intuitive example. Consider two models: one for age prediction and another for sentiment classification. If a user wants to recognize hair color, which model is better suited? With output spaces differing in structure and meaning, direct alignment and comparison become challenging. Moreover, if the models vary in quality, which should the system recommend? To protect data privacy, real user data cannot be submitted online for evaluation, and requiring users to evaluate multiple models locally and provide feedback is impractical. Next, we present a solution to address both challenges simultaneously.

The capability of a model $h$ is characterized by the relationship between the input $\mathbf{x}$ and its prediction $h(\mathbf{x})$, formalized as $p(h(\mathbf{x})|\mathbf{x})$. It consists of two components: task semantics and model quality. The former is determined by the input and output space, while the latter arises from the alignment between the predicted value $\hat{y}$ and the true label $y$. Meanwhile, the requirements of a user task can be formalized as the conditional probability $p_u(y|\mathbf{x})$, reflecting the required capability to solve it.

In this paper, parameter vectors are categorized into two types based on their functionality: **model vectors** to represent the model capability, and **task vectors** to represent the user task requirements. For the model $h$ trained on task $(\mathcal{L}_t, \mathcal{D}_t)$, the corresponding model vector can be constructed through fine-tuning the pre-trained model $f$ to represent its capability, $p(h(\mathbf{x})|\mathbf{x})$. In some cases, the output space of the pre-trained model $f$ is not $\mathcal{Y}$ but rather a feature space $\mathcal{Z} \subseteq \mathbb{R}^O$. And, it is common practice to bridge the feature space $\mathcal{Z}$ to the output space $\mathcal{Y}$ using a function $g : \mathcal{Z} \rightarrow \mathcal{Y}$, such that $g \circ f : \mathcal{X} \rightarrow \mathcal{Y}$. Typically, the function $g$ can be constructed without training via *Prompt-Tuning* (Lester et al., 2021). In general, we can obtain the model vector $\boldsymbol{\tau}_h$ for the model $h$ through

$$\boldsymbol{\tau}_h = \arg\min_{\boldsymbol{\tau}} \sum_{(\mathbf{x},y) \in \mathcal{D}_t} \mathcal{L}_t(g_t \circ f(\mathbf{x}, \theta_0 + \boldsymbol{\tau}), h(\mathbf{x})). \tag{1}$$

And for the task vector $\boldsymbol{\tau}_u$, the prediction $h(\mathbf{x})$ should be replaced with the true label $y$, reflecting the required capability to solve the user task. Then, the alignment between the model capability and the user task requirements can be assessed by the PAVE similarity

$$\cos(\boldsymbol{\tau}_h, \boldsymbol{\tau}_u) = \mathbf{Similarity}(p(h(\mathbf{x})|\mathbf{x}), p_u(y|\mathbf{x})). \tag{2}$$

Accordingly, we identify the learnware $(h, \boldsymbol{\tau}_h)$ to solve the user task with specification $\boldsymbol{\tau}_u$ if $\cos(\boldsymbol{\tau}_h, \boldsymbol{\tau}_u)$ is high. In practice, we identify the one with the highest similarity.

A model is identified only if its capability $p(h(\mathbf{x})|\mathbf{x})$ aligns closely with the user task requirements $p_u(y|\mathbf{x})$. Besides, if a model is of poor quality, its capability $p(h(\mathbf{x})|\mathbf{x})$ deviates significantly from the true task semantics $p(y|\mathbf{x})$ on the training dataset. Therefore, even if the training dataset accurately reflects the user task–i.e., $p(y|\mathbf{x}) = p_u(y|\mathbf{x})$–a low-quality model will still not be identified.

## 3.2 THEORETICAL CONNECTION BETWEEN PAVE AND RKME

To clarify the relationship between PAVE and prior specifications–specifically RKME (see Appendix F.1 for details)–we note that, under the NTK assumption, a parameter vector can be viewed as an accumulation of gradients over the task data. This implies that the inner product between PAVEs corresponds to a kernel function over the tasks' sample spaces, with each parameter vector representing a kernel mean embedding. Consequently, PAVE similarity and the maximum mean discrepancy (MMD) employed by RKME exhibit order consistency in evaluating model-task alignment.

Using gradient descent, the model update vector $\boldsymbol{\tau}_h$ can be expressed as an accumulation of gradients:

$$\nabla_{\boldsymbol{\tau}} \mathcal{L}_t(y_i', \hat{y}_i) = \nabla_{\boldsymbol{\tau}} \mathcal{L}_t(g_t \circ f(\mathbf{x}_i, \theta_0 + \boldsymbol{\tau}), h(\mathbf{x}_i)) = \nabla_{y_i'} \mathcal{L}_t \, \nabla_z g_t \, \nabla_\theta f(\mathbf{x}_i, \theta_0) \in \mathbb{R}^{|\theta|}, \tag{3}$$

where $y_i' = g_t \circ f(\mathbf{x}_i, \theta_0 + \boldsymbol{\tau})$ and $\mathbb{R}^{|\theta|}$ is the parameter space. The second equality in Equation (3) holds if the fine-tuning process satisfies the neural tangent kernel (NTK) behavior (Lee et al., 2019), where the gradient function evolves with accumulated updates but remains approximately fixed under linearized dynamics. The same formulation applies to the task vector $\boldsymbol{\tau}_u$, with $\hat{y}_j$ replaced by $y_j$.

**Lemma 1.** *Suppose the fine-tuning process lies in the neural tangent kernel regime. Then the parameter update vector is the accumulation of gradients*

$$\nabla_{\boldsymbol{\tau}} \mathcal{L}_t(y_i', \hat{y}_i) = \nabla_{y_i'} \mathcal{L}_t \, \nabla_z g_t \, \nabla_\theta f(\mathbf{x}_i, \theta_0) \in \mathbb{R}^{|\theta|}. \tag{4}$$

*Moreover, the inner product between samples from the training task $(\mathcal{D}_t, \mathcal{L}_t)$ of model $h$ and from the user task $(\mathcal{D}_u, \mathcal{L}_u)$ defines an implicit kernel function $\tilde{k}_{f,\mathcal{L},g}$:*

$$\langle \nabla_{\boldsymbol{\tau}} \mathcal{L}_t(y_i', \hat{y}_i), \nabla_{\boldsymbol{\tau}} \mathcal{L}_u(y_j', y_j) \rangle = \sum_{r,s} (\tilde{K}_f \circ \tilde{K}_{\mathcal{L},g})_{rs} = \tilde{k}_{f,\mathcal{L},g}(\mathbf{x}_i, h(\mathbf{x}_i), \mathbf{x}_j, y_j), \tag{5}$$

*where the empirical neural tangent kernel $\tilde{K}_f(\mathbf{x}_i, \mathbf{x}_j) = \nabla_\theta f(\mathbf{x}_i, \theta_0) \nabla_\theta^\top f(\mathbf{x}_j, \theta_0)$ (Mohamadi et al., 2023), and $\tilde{K}_{\mathcal{L},g}(h(\mathbf{x}_j), y_j) = (\nabla_{y_i'} \mathcal{L}_t \nabla_z g_t)^\top (\nabla_{y_j'} \mathcal{L}_u \nabla_z g_u)$.*

The component $\tilde{K}_{\mathcal{L},g}$ is dynamically constructed from the task semantics induced by the loss functions $\mathcal{L}_t, \mathcal{L}_u$ and mappings $g_t, g_u$. It acts as a consistency weight matrix applied element-wise to $\tilde{K}_f$, which encourages similarity between pairs of features sharing the same correlation with solving the task and penalizes the similarity of others. Therefore, the implicit kernel function $\tilde{k}_{f,\mathcal{L},g}$ in Lemma 1 jointly encodes input data $\mathbf{x}$, model predictions $h(\mathbf{x})$, and task semantics (Domingos, 2020). The more consistent the input–output relationships between two samples, the larger the kernel value. The detailed derivation of the concrete construction of the kernel $\tilde{k}_{f,\mathcal{L},g}$ is given in Appendix A.

**Lemma 2.** *For the model vector $\boldsymbol{\tau}_h$ and the task vector $\boldsymbol{\tau}_u$, their inner product can be expressed as*

$$\langle \boldsymbol{\tau}_h, \boldsymbol{\tau}_u \rangle = \sum_{(\mathbf{x}_i, y_i) \in \mathcal{D}_t} \sum_{(\mathbf{x}_j, y_j) \in \mathcal{D}_u} \tilde{k}_{f,\mathcal{L},g}(\mathbf{x}_i, h(\mathbf{x}_i), \mathbf{x}_j, y_j). \tag{6}$$

*With normalization constant $Z$ chosen such that $\langle \boldsymbol{\tau}, \boldsymbol{\tau} \rangle = 1$, the PAVE $\boldsymbol{\tau}$ can be expressed in a form analogous to an empirical kernel mean embedding (KME), up to a rescaling factor:*

$$\boldsymbol{\tau} = \frac{1}{Z} \sum_{i=1}^{m} \tilde{k}_{f,\mathcal{L},g}(\mathbf{z}_i, \cdot), \text{ where } \mathbf{z}_i = (\mathbf{x}_i, \mathbf{y}_i'). \tag{7}$$

To eliminate the bias introduced by dataset size and optimization trajectory length, we normalize the gradient sum when defining $\boldsymbol{\tau}$. With the definition of $\boldsymbol{\tau}$ in place, we are now ready to present an important result showing the order-consistency between PAVE similarity and MMD.

**Theorem 3.** *Suppose the fine-tuning process operates in the neural tangent kernel (NTK) regime, under which the parameter vector admits an interpretation as a kernel mean embedding. Then, for any user task, the similarity between PAVEs and the MMD between the corresponding distributions with respect to the kernel $\tilde{k}_{f,\mathcal{L},g}$ are **order-consistent**, inducing the same preference between models.*

Building on Lemmas 1 and 2, Theorem 3 is a natural consequence, as both measures clearly induce the same ordering. From the PAVE perspective, this ordering reflects the similarity between the directions of two fine-tuning trajectories in function space, whereas from the MMD perspective, it captures the distance between the corresponding distributions in RKHS (Gretton et al., 2012).

### 3.3 APPROXIMATION OF PARAMETER VECTOR SIMILARITY

The weight space of modern pre-trained models is typically in the hundreds of millions, making direct construction and comparison of parameter vectors derived from the weight space impractical. We propose a method to approximate parameter vectors in a low-rank space, reducing their size to less than 1% of the pre-trained model's parameters—often fewer than 1M—thereby significantly enhancing efficiency in construction, comparison, and storage. Furthermore, we provide a theoretical guarantee that our approach preserves similarity with high probability, as illustrated in Figure 3.

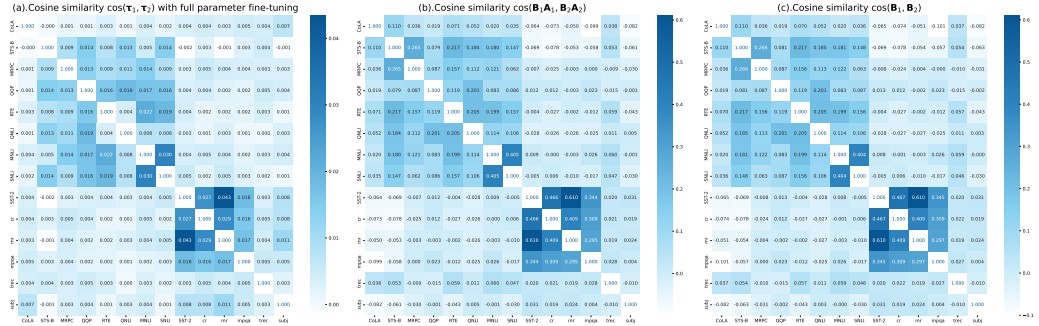

Figure 3: **Consistency of the cosine similarity between parameter vectors**, with the diagonal colors omitted to enhance visual contrast. (a) shows the exact similarity of the parameter vectors with full parameter fine-tuning. (b) shows the approximating similarity in the low-rank space after expanding parameter vectors to $(\mathbf{BA})_{m \times n}$ in full size. (c) shows the approximating similarity computed using $\mathbf{B}$ alone for improved storage and computational efficiency, which is the method we propose.

Specifically, we approximate the parameter vector $\boldsymbol{\tau}$ in a LoRA (Hu et al., 2022) manner that

$$\boldsymbol{\tau} \approx \tilde{\boldsymbol{\tau}} \triangleq [\Delta \mathbf{W}^1 \; \Delta \mathbf{W}^2 \; \ldots \; \Delta \mathbf{W}^L] = [(\mathbf{B}^1 \mathbf{A}^1) \, (\mathbf{B}^2 \mathbf{A}^2) \; \ldots \; (\mathbf{B}^L \mathbf{A}^L)] \triangleq \mathbf{BA}, \qquad (8)$$

where the pre-trained model consists of $L$ weight matrices $\{\mathbf{W}^i\}_L$. To simplify notations, we assume that $\mathbf{A} \triangleq [\mathbf{A}^1 \; \mathbf{A}^2 \; \ldots \; \mathbf{A}^L]$ and $\mathbf{B}$ likewise, where $\mathbf{A}$ is initialized randomly and $\mathbf{B}$ is initialized with zeros. As an improvement to Equation (1), we replace the parameter vector $\boldsymbol{\tau}$ with a low-rank approximated $\mathbf{BA}$, leading to a significant reduction in construction and storage overhead.

With constructed model and task vectors, it's still challenging to compare their similarity efficiently. If we expand the metrics $\mathbf{B}'_{m \times r}$ and $\mathbf{A}'_{r \times n}$ to $(\mathbf{B}'\mathbf{A}')_{m \times n}$ in full size, it costs the same computational overhead for computing cosine similarity as without approximation. To avoid this, we propose to approximate $\cos(\tilde{\boldsymbol{\tau}}_1, \tilde{\boldsymbol{\tau}}_2)$ by computing $\cos(\mathbf{B}_1, \mathbf{B}_2)$ only, where the parameter vectors are implicitly flattened for the inner product and cosine similarity calculations. We gonna give some key ideas here and the rigorous proof is presented in Appendix B. First, it's easy to verify that in expectation

$$\mathbb{E}[\mathbf{A}^l (\mathbf{A}^l)^\top] = s\mathbf{I} \quad \text{and} \quad \mathbb{E}[\langle \tilde{\boldsymbol{\tau}}_1, \tilde{\boldsymbol{\tau}}_2 \rangle] = \mathbb{E}[\langle \mathbf{B}_1 \mathbf{A}, \mathbf{B}_2 \mathbf{A} \rangle] = s \langle \mathbf{B}_1, \mathbf{B}_2 \rangle, \qquad (9)$$

where $\mathbf{A} \sim \mathcal{U}(-\sqrt{3s/n}, +\sqrt{3s/n})$ element-wisely with a constant factor $s$. Note that it is Kaiming initialization (He et al., 2015) with proper $s$. Then, for the cosine similarity, we can prove that

$$\cos(\tilde{\boldsymbol{\tau}}_1, \tilde{\boldsymbol{\tau}}_2) \approx \cos(\mathbf{B}_1, \mathbf{B}_2). \qquad (10)$$

And more details are presented in Theorem 4. In summary, the exact parameter vector similarity $\cos(\boldsymbol{\tau}_1, \boldsymbol{\tau}_2)$ is equal to $\cos(\mathbf{B}_1, \mathbf{B}_2)$ in the low-rank space approximately.

## 4 EXPERIMENTS

In this section, we evaluate our method on natural language processing and computer vision tasks. We first outline the experimental setup, including the dataset, compared methods, and experiment configuration. Our results show that identified learnwares outperform fine-tuning pre-trained models on user tasks and surpass the previous methods, particularly in solving tasks beyond the original functionality of the learnwares. Furthermore, ablation studies confirm the effectiveness and necessity of our method in preserving model quality while maintaining computational efficiency.

### 4.1 EXPERIMENTAL SETUP

We give a brief description of the evaluation dataset, the compared methods, and the experiment configuration. And, more specific information can be found in Appendix C.

**Datasets.** We evaluate our proposed method on 15 classic natural language processing datasets, 9 medical LLM benchmarks and 12 computer vision datasets, 9 medical SFT datasts and 9 medical

Table 1: Performances on NLP datasets beyond the original functionality.

| User Task | | Finetune Pre-trained Models | | | | Identify Learnware | | | |
|---|---|---|---|---|---|---|---|---|---|
| Dataset | Metric | BERT-B | BERT-L | RoBERTa-B | RoBERTa-L | Random | RKME | PAVE$^\diamond$ | Oracle |
| CoLA | Mcc | 0.049 | 0.012 | 0.086 | 0.064 | 0.095 | 0.083 | **0.100** | 0.162 |
| MNLI | Acc | 0.498 | 0.519 | 0.594 | 0.646 | 0.582 | 0.628 | **0.677** | 0.677 |
| MRPC | Acc | 0.537 | 0.600 | 0.662 | 0.703 | 0.700 | 0.718 | **0.782** | 0.782 |
| MRPC | F1 | 0.609 | 0.694 | 0.749 | 0.776 | 0.782 | 0.799 | **0.854** | 0.854 |
| QNLI | Acc | 0.552 | 0.542 | 0.636 | 0.682 | 0.643 | 0.580 | **0.732** | 0.749 |
| QQP | Acc | 0.574 | 0.512 | 0.653 | 0.664 | 0.646 | **0.696** | 0.631 | 0.715 |
| QQP | F1 | 0.542 | 0.558 | 0.648 | 0.664 | 0.637 | **0.686** | 0.657 | 0.686 |
| RTE | Acc | 0.527 | 0.560 | 0.523 | **0.686** | 0.596 | 0.635 | 0.560 | 0.668 |
| SNLI | Acc | 0.515 | 0.575 | 0.676 | 0.753 | 0.671 | 0.702 | **0.755** | 0.755 |
| SST-2 | Acc | 0.811 | 0.864 | 0.889 | **0.937** | 0.872 | 0.896 | 0.896 | 0.896 |
| STS-B | Pcc | 0.332 | 0.536 | **0.740** | 0.620 | 0.603 | 0.440 | 0.718 | 0.778 |
| STS-B | Scc | 0.312 | 0.529 | **0.734** | 0.615 | 0.593 | 0.424 | 0.722 | 0.781 |
| WNLI | Acc | 0.408 | 0.408 | 0.465 | 0.423 | 0.458 | 0.437 | **0.479** | 0.521 |
| cr | Acc | 0.873 | 0.881 | 0.906 | **0.920** | 0.886 | 0.888 | 0.888 | 0.902 |
| mpqa | Acc | 0.789 | 0.775 | 0.822 | 0.812 | 0.810 | 0.814 | **0.826** | 0.838 |
| mr | Acc | 0.757 | 0.802 | 0.849 | **0.877** | 0.828 | 0.850 | 0.850 | 0.850 |
| subj | Acc | 0.858 | 0.883 | **0.910** | 0.909 | 0.894 | 0.885 | 0.885 | 0.911 |
| trec | Acc | 0.752 | 0.728 | 0.740 | **0.824** | 0.732 | 0.702 | 0.750 | 0.770 |
| Avg.($\uparrow$) | | 0.572 | 0.610 | 0.682 | 0.699 | 0.668 | 0.659 | **0.709** | 0.739 |
| Avg. rank($\downarrow$) | | 6.389 | 6.000 | 3.333 | 2.444 | 4.056 | 3.389 | **2.111** | - |
| PAVE$^\diamond$ (win/tie/loss) | | 17/0/1 | 17/1/0 | 13/0/5 | 10/0/8 | 15/0/3 | 12/3/3 | - | 0/6/12 |

LLM benchmarks. The natural language processing tasks are mainly from the GLUE benchmark (Wang et al., 2019a) and some other datasets. And the 12 computer vision tasks incorporate diverse task semantics, such as satellite images in EuroSAT (Helber et al., 2019), traffic signs in GTSRB (Stallkamp et al., 2012) and scene recognition in SUN397 (Xiao et al., 2010). And the medical LLM benchmarks come from the Open Medical-LLM Leaderboard(Pal et al., 2024), following the setup in (Tan et al., 2025). The details of the datasets are described in Appendix C.1.

**Compared methods.** We evaluate our method, denoted as PAVE, against famous pre-trained models fine-tuned for the user task, including RoBERTa (Liu et al., 2019), BERT (Devlin et al., 2019), and CLIP (Radford et al., 2021). Additionally, we include RKME (Tan et al., 2024b) from previous studies as a compared method and introduce several baseline methods. Specifically, "Random" refers to the random identification of learnwares, representing the average capability of the learnwares, while "Oracle" denotes the selection of learnwares that perform best on the user task, representing the optimal capability. A specific description of the compared methods can be found in Appendix C.2.

**Experiment configuration** of learnware set and user tasks. Each dataset consists of two subsets: a training dataset and a test dataset. In this experiment, the training dataset is further divided into two parts: the dataset $\mathcal{D}_t$ for developers to constructe the learnwares and the k-shot dataset $\mathcal{D}_u$ as part of the user task $(\mathcal{D}_u, \mathcal{L}_u)$. To construct the learnware set, the developers trained the models on the dataset $\mathcal{D}_t$, paired them with the corresponding model vectors as specifications, and added them to the set. Specifically, for NLP and CV tasks, each dataset provided two versions of learnware: a well-trained version and a corrupted version (consisting of low-quality models). And for models of the same type, whether corrupted or not, they are trained using identical hyper-parameters (e.g., learning rate, number of epochs) to ensure generality. For medical LLM tasks, we only use high-quality models as the inherent output degeneracy (e.g., self-repetition) triggered by destructive fine-tuning precludes any meaningful evaluation; thus, model vectors are directly obtained by fitting the training data to characterize model capabilities. Additionally, both the construction of task vectors and the fine-tuning of pre-trained models on the user tasks are performed within the few-shot learning framework. Finally, the identified learnware is evaluated on the corresponding test dataset. The detailed process of learnware training and the hyper-parameter configurations can be found in Appendix C.3.

## 4.2 PERFORMANCES

We show that learnwares identified by our proposed method outperforms fine-tuned pre-trained models on user tasks, which is the most popular approach for solving new deep learning tasks in the past. Moreover, our proposed method significantly improves upon the previous RKME method. We

also verified that in many tasks, our proposed method can identify learnware that efficiently solve tasks beyond their original function and exhibit strong generalization capabilities.

### 4.2.1 ALIGNING MODELS WITH USER TASKS

This experiment confirms that PAVE specification can identify the model from the candidate set that best aligns with the user task requirements. And to ensure objectivity and completeness, we evaluated PAVE on both natural language processing and computer vision tasks.

**Classic NLP.** The results of our evaluation are presented in Table A5 and Figure A7, demonstrating the advantages of PAVE. And the evaluation metrics follow the GLUE benchmark evaluation settings. The performance of PAVE specification significantly surpasses the random selection and is close to the optimal capability, demonstrating better results than RKME. Remarkably, PAVE is tied with Oracle and identifies the optimal learnware on 14 out of 18 user tasks.

**Medical LLMs.** The evaluation results on Open Medical LLM Leaderboard (Pal et al., 2024; Singhal et al., 2022) are presented in Table A4, demonstrating the effectiveness of PAVE specification in identifying large language models. We compare our approach with three popular LLMs and other baseline methods, showing that our method offers a clear overall advantage. Additionally, a visual illustration of the identification results is provided in Figure A6 for further insights.

**Computer vision.** The evaluation results are presented in Table A6 and Figure A8. PAVE effectively identifies the most proficient candidate learnware for the user task. Furthermore, it demonstrates clear advantages over fine-tuning pre-trained models and previous RKME methods.

### 4.2.2 BEYOND THE ORIGINAL FUNCTIONALITY

This experiment demonstrates that the identified learnwares can be utilized beyond their original functionality, meaning that models can be repurposed to assist with tasks not originally intended by their developers. For each user task, we remove all learnwares in the candidate set that have been trained on the corresponding training dataset, presenting results in Table 1 and Figure A9, where "PAVE$^\diamond$" represents the performance after fine-tuning the identified learnware on the user task.

It shows that fine-tuning the identified learnware achieves better performance compared to fine-tuning pre-trained models. This finding suggests that learnwares can effectively tackle tasks beyond their original functionality, even outperforming popular pre-trained models. One possible explanation is that the learnware candidate set, as a whole, encompasses a broader range of capabilities than a single pre-trained model, as shown in Figure 2. Furthermore, the PAVE specification outperforms the previous method, RKME, by an average of **7.59%** in Table 1.

### 4.3 ABLATION STUDY

We carry out ablation studies to demonstrate the necessity of the components in our method. First, we verify the necessity of sketching the model capability $p(\hat{y}|\mathbf{x})$ for capturing model quality, instead of the data distribution $p(y|\mathbf{x})$. Second, we show that approximating the parameter vectors in the low-rank space truly leads to great speedups and is almost harmless for the performance.

### 4.3.1 MODEL QUALITY

To verify that fitting the model capability $p(\hat{y}|\mathbf{x})$ instead of the data distribution $p(y|\mathbf{x})$ is necessary for filtering out low-quality learnwares, we construct a corrupted version of each learnware that has not been trained on the training dataset and include them in the candidate set. The results of the ablation experiments are presented in Tables 2 and A8, where "PAVE$^\clubsuit$" represents the performance of model vectors derived from fitting the data distribution $p(y|\mathbf{x})$, serving as the ablation variant. Notably, the ablation variant "PAVE$^\clubsuit$" performs significantly worse than PAVE, highlighting the necessity of fitting the model capability $p(\hat{y}|\mathbf{x})$ when the model's quality is not guaranteed.

### 4.3.2 PARAMETER VECTOR APPROXIMATION

We confirm that approximating parameter vectors in low-rank spaces significantly improves storage and computational efficiency. The computational overhead comprises two primary components:

Table 2: Performances on computer vision datasets with corrupted learnwares.

| User Task | Finetune Pre-trained Models | | | Identify Learnware | | | |
| | ResNet-152 | ViT-B-32 | ViT-L-14 | Random | PAVE♣ | PAVE | Oracle |
| Dataset | | | | | | | |
|---|---|---|---|---|---|---|---|
| CIFAR10 | 0.100 | 0.899 | **0.956** | 0.839 | 0.894 | 0.890 | 0.979 |
| CIFAR100 | 0.008 | 0.663 | 0.777 | 0.585 | 0.764 | **0.885** | 0.885 |
| Cars | 0.006 | 0.604 | **0.784** | 0.545 | 0.639 | 0.680 | 0.680 |
| DTD | 0.022 | 0.440 | 0.556 | 0.422 | 0.568 | **0.698** | 0.698 |
| EuroSAT | 0.161 | 0.541 | 0.536 | 0.465 | 0.742 | **0.991** | 0.991 |
| Food101 | 0.008 | 0.828 | **0.924** | 0.762 | 0.856 | 0.886 | 0.886 |
| GTSRB | 0.030 | 0.330 | 0.511 | 0.338 | 0.656 | **0.987** | 0.987 |
| MNIST | 0.117 | 0.503 | 0.764 | 0.517 | 0.740 | **0.997** | 0.997 |
| RESISC45 | 0.019 | 0.611 | 0.713 | 0.565 | 0.770 | **0.934** | 0.934 |
| STL10 | 0.060 | 0.973 | **0.994** | 0.949 | 0.978 | 0.985 | 0.985 |
| SUN397 | 0.003 | 0.648 | 0.694 | 0.611 | 0.686 | **0.739** | 0.739 |
| SVHN | 0.096 | 0.329 | 0.584 | 0.357 | 0.643 | **0.970** | 0.970 |
| Avg.($\uparrow$) | 0.052 | 0.614 | 0.733 | 0.580 | 0.745 | **0.887** | 0.894 |
| Avg. rank($\downarrow$) | 6.000 | 4.000 | 2.167 | 4.750 | 2.583 | **1.500** | - |
| PAVE (win/tie/loss) | 12/0/0 | 11/0/1 | 8/0/4 | 12/0/0 | 11/0/1 | - | 0/11/1 |
| PAVE♣ (win/tie/loss) | 12/0/0 | 11/0/1 | 5/0/7 | 12/0/0 | - | 1/0/11 | 0/0/12 |

constructing the specifications and computing their similarity. Similarly, storage overhead refers to the space required to store them. In Figures 3, A13 and A14, the results demonstrate that the approximation preserves the relative relationships in terms of cosine similarity. Table A9 summarizes the computational and storage overheads associated with each method, along with their performance. These results underscore the benefits of approximating parameter vectors in low-rank spaces.

## 4.4 ANALYSIS

In order to gain a deeper understanding of PAVE specification, we perform some further analysis in Appendix E. We perform a Hierarchical Linear Regression (Woltman et al., 2012) to show the significant positive correlation between similarity and performance. Moreover, we apply Exact Multinomial Test (Read and Cressie, 2012) to verify that PAVE is able to identify the helpful learnware with statistical significance. These findings support the effectiveness of PAVE strongly.

## 5 RELATED WORK

This section reviews related topics and illustrates their relevance to our work. First, our work contributes to *Learnware* paradigm by enhancing its identification for deep learning models and characterizes model capability in a more generalizable and efficient way. Second, we evaluate the model's *Transferability* to user tasks without unscalable per-model evaluations while ensuring data privacy. We also advance the understanding of *Parameter Vector* by offering insights into the relationship between changes in model parameters and the model capabilities.

**Learnware.** The *learnware* paradigm (Zhou, 2016; Zhou and Tan, 2024) leverages specifications to identify and reassemble multiple heterogeneous models for emergent capabilities without accessing original training data. A key method for identifying suitable learnware is the Reduced Kernel Mean Embedding (RKME) specification (Zhou and Tan, 2024), which matches the user's data distribution with the model's original data distribution. Its unsupervised version has been demonstrated effective in various tasks (Wu et al., 2023). Recently, there have been various subsequent works that reinforce the feasibility of the learnware paradigm. Lei et al. (2024) theoretically prove its privacy-preserving ability of the learnware paradigm for developers' raw data. Liu et al. (2024) introduce the Evolvable Learnware Specification with Index (ELSI), which increasingly improves model capability characterization as the number of learnware increases. Tan et al. (2023; 2024a) propose learnware search and reuse algorithms to leverage learnwares from heterogeneous feature spaces, and Guo et al. (2023) try to utilize learnwares from heterogeneous label spaces. Xie et al. (2023) explore the methods for identifying learnware efficiently by constructing anchor learnwares. In order to facilitate research

on learnware, Beimingwu learnware research platform has been released (Tan et al., 2024b) and have further contributed to related research in this area. Tan et al. (2025) first apply the learnware paradigm to SLMs with parameter vector specification, demonstrating performance superior to LLMs across specialized tasks. Our work contributes to Learnware by presenting a comprehensive study of PAVE specification and enhancing learnware identification for deep models by characterizing model capability in the parameter space.

**Transferability.** The transferability of a pre-trained model (PTM) is critical for its effective adaptation from a source to a target task. Many methods estimate transferability via a forward pass on the target task, leveraging different metrics to approximate performance. Notable approaches include NCE (Tran et al., 2019), LEEP (Nguyen et al., 2020), LogME (You et al., 2021), PACTran (Ding et al., 2022), and TransRate (Huang et al., 2022), which estimate negative conditional entropy, log expectation, marginalized likelihood, PAC-Bayesian bounds, and mutual information, respectively. While these methods reduce the need for full fine-tuning, they incur high computational costs due to the complexity of modern PTMs and are limited by PTM-specific embeddings. Additionally, transferability depends on the relatedness between pre-training and downstream tasks (Ben-David and Schuller, 2003; Wang et al., 2019b; Lu et al., 2021; Lin et al., 2024). Various approaches assess task relatedness, including full fine-tuning (Zamir et al., 2018), example-based graphs (Dwivedi and Roig, 2019), optimal transport between label spaces (Lu et al., 2022), and representation-level similarities (Dwivedi et al., 2022). Recently, Model Spider (Zhang et al., 2023) leveraged historical data to learn task and PTM similarities, enhancing model selection. However, these methods generally require direct access to user data, and computational constraints hinder per-model evaluations on user devices. These challenges can be addressed naturally based on specifications in learnware paradigm, advancing transferability research by leveraging broader real-world model platform.

**Parameter Vector.** The change in model parameters, which we term the parameter vector, has been explored in gradient matching and, more recently, as task vectors. Gradient matching is used for distilling large datasets into smaller synthetic ones (Zhao et al., 2021; Cazenavette et al., 2022) or domain generalization (Shi et al., 2022). More recently, task vectors have been increasingly used to merge models for multi-task learning (Ilharco et al., 2023; Ortiz-Jiménez et al., 2023; Tang et al., 2024a; Yadav et al., 2023; Yang et al., 2024; Xu et al., 2025a; Zhang and Zhou, 2025; Yoshida et al., 2025; Jin et al., 2025; Gargiulo et al., 2025). Besides, Li et al. (2025) analyze the generalization ability of task vectors and demonstrate the feasibility of low-rank approximation, while Rinaldi et al. (2025) investigate how to use task vectors to transfer fine-tuned knowledge to new backbones without retraining. Further studies have proposed advanced merging strategies to avoid parameter conflicts (Lai et al., 2025) or measured shifts in internal activations for model selection (Xu et al., 2025b). This research primarily focuses on combining model capabilities. In contrast, our work explores a different direction: we investigate the relationship between parameter vector and model capability similarities, not for merging models, but for identifying the best models for a specific user task.

## 6 CONCLUSION AND DISCUSSIONS

In this paper, we provide a comprehensive study of PAVE specifications, which sketch model capability and task requirements with parameter vectors to efficiently capture reusability relations, to identify and reuse high-quality learnware that effectively solves the user task. Theoretical analysis and extensive experiments verify its effectiveness, and shed light on its underlying mechanism. The method outperforms user-fine-tuned pre-trained models, exhibiting remarkable generalization capabilities and successfully solving tasks beyond their original functionality. Furthermore, the approach demonstrates robustness by selecting the good learnwares even from repositories of corrupted quality. Finally, we verify that approximating parameter vectors in the low-rank space enhances both storage and computational efficiency with minimal impact on performance. These findings show the parameter vector's potential in the learnware paradigm, suggesting future work on efficient generation for resource-constrained settings (Tang et al., 2024b).

## ACKNOWLEDGEMENTS

This work was supported by Jiangsu Science Foundation Leading-edge Technology Program (BK20232003) and Collaborative Innovation Center of Novel Software Technology and Industrial-

ization. Zhi-Hao Tan was supported by JiangsuSF (BK20251215) and the Postdoctoral Fellowship Program of CPSF under Grant Number GZB20250396. This work was also supported by Fundamental and Interdisciplinary Disciplines Breakthrough Plan of the Ministry of Education of China (No. JYB2025XDXM118). We thank Tian-Zuo Wang, Han-Jia Ye, Lan-Zhe Guo, Peng Tan, Jia-Wei Shan, Hao-Yi Lei and Xin-Yu Zhang for their helpful discussions. We are also grateful to the anonymous reviewers for their helpful comments.

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

# A   DETAILS ON THE CONNECTION BETWEEN PAVE AND RKME

In this section, we present a detailed discussion on why the cosine similarity between parameter vectors effectively captures the alignment between model capabilities and task requirements, as illustrated in Equation (2). Furthermore, we demonstrate that this similarity corresponds to an implicit kernel function grounded in task semantics and the empirical Neural Tangent Kernel (NTK).

The model vector $\boldsymbol{\tau}_h$ with respect to the training task $(\mathcal{D}_t, \mathcal{L}_t)$ and model $h$ can be obtained by solving this optimization formula:

$$\boldsymbol{\tau}_h = \arg\min_{\boldsymbol{\tau}} \sum_{(\mathbf{x},y)\in\mathcal{D}_t} \mathcal{L}(g_t \circ f(\mathbf{x}, \theta_0 + \boldsymbol{\tau}), h(\mathbf{x})). \tag{11}$$

When we optimize it with gradient descent, the model vector we obtain is the accumulation of the gradients of the loss function $\mathcal{L}(g_t \circ f(\mathbf{x}, \theta_0 + \boldsymbol{\tau}), h(\mathbf{x}))$ with respect to $\boldsymbol{\tau}$. Accordingly, the gradient of the loss function on the sample $(\mathbf{x}_i, y_i)$ is

$$\nabla_{\boldsymbol{\tau}} \mathcal{L}_t(y_i', \hat{y}_i) = \nabla_{\boldsymbol{\tau}} \mathcal{L}_t(g_t \circ f(\mathbf{x}_i, \theta_0 + \boldsymbol{\tau}), h(\mathbf{x}_i)) = \nabla_{y_i'} \mathcal{L}_t \nabla_z g_t \nabla_\theta f(\mathbf{x}_i, \theta_0) \in \mathbb{R}^{|\theta|}, \tag{12}$$

where $y_i' = g \circ f(\mathbf{x}_i, \theta_0 + \boldsymbol{\tau})$ and $\mathbb{R}^{|\theta|}$ is the weight space. Note that the second equality in Equation (12) holds if the fine-tuning process satisfies the neural tangent kernel behaviour (Lee et al., 2019). And, the model vector $\boldsymbol{\tau}_h$ is essentially the accumulation of gradients

$$\boldsymbol{\tau}_h = \sum_{(\mathbf{x},y)\in\mathcal{D}_t} \nabla_{\boldsymbol{\tau}} \mathcal{L}(y_i', h(\mathbf{x}_i)) = \sum_{(\mathbf{x},y)\in\mathcal{D}_t} \nabla_{y_i'} \mathcal{L} \nabla_z g_t \nabla_\theta f(\mathbf{x}_i, \theta_0). \tag{13}$$

This also holds for the task vector $\boldsymbol{\tau}_u$, with the user task $(\mathcal{D}_u, \mathcal{L}_u)$ and the true label $y$. Then, the inner product between model vector and task vector is defined as

$$\begin{aligned}
\langle \boldsymbol{\tau}_h, \boldsymbol{\tau}_u \rangle &= \sum_{(\mathbf{x}_i,y_i)\in\mathcal{D}_t} \sum_{(\mathbf{x}_j,y_j)\in\mathcal{D}_u} \nabla_{\boldsymbol{\tau}} \mathcal{L}_t(y_i', h(\mathbf{x}_i)) \nabla_{\boldsymbol{\tau}}^\top \mathcal{L}_u(y_j', y_j) \\
&= \sum_{(\mathbf{x}_i,y_i)\in\mathcal{D}_t} \sum_{(\mathbf{x}_j,y_j)\in\mathcal{D}_u} \nabla_{y_i'} \mathcal{L}_t \nabla_z g_t \nabla_\theta f(\mathbf{x}_i, \theta_0) \nabla_\theta^\top f(\mathbf{x}_j, \theta_0) \nabla_z^\top g_u \nabla_{y_j'}^\top \mathcal{L}_u.
\end{aligned} \tag{14}$$

We can define the similarity in the summation equation as a kernel function $\tilde{k}_{f,\mathcal{L},g}$ that

$$\begin{aligned}
\tilde{k}_{f,\mathcal{L},g}(\mathbf{x}_i, h(\mathbf{x}_i), \mathbf{x}_j, y_j) &= \nabla_{\boldsymbol{\tau}} \mathcal{L}_t(y_i', h(\mathbf{x}_i)) \nabla_{\boldsymbol{\tau}}^\top \mathcal{L}_u(y_j', y_j) \\
&= \left(\nabla_{y_i'} \mathcal{L}_t \nabla_z g_t\right) \tilde{K}_f(\mathbf{x}_i, \mathbf{x}_j) \left(\nabla_{y_j'} \mathcal{L}_u \nabla_z g_u\right)^\top \in \mathbb{R}.
\end{aligned} \tag{15}$$

Note that $\tilde{K}_f(\mathbf{x}_i, \mathbf{x}_j)$ is the empirical neural tangent kernel (Mohamadi et al., 2023)

$$\tilde{K}_f(\mathbf{x}_i, \mathbf{x}_j) = \nabla_\theta f(\mathbf{x}_i, \theta_0) \nabla_\theta^\top f(\mathbf{x}_j, \theta_0) \in \mathbb{R}^{O\times O}, \tag{16}$$

where $O$ denotes the dimension of the feature space $\mathcal{Z}$.

To understand this kernel function $\tilde{k}_{f,\mathcal{L},g}(\mathbf{x}_i, h(\mathbf{x}_i), \mathbf{x}_j, y_j)$ even further, we can decompose it into two parts $\tilde{K}_f(\mathbf{x}_i, \mathbf{x}_j)$ and $\tilde{K}_{\mathcal{L},g}(h(\mathbf{x}_j), y_j)$ with some trace tricks:

$$\begin{aligned}
\tilde{k}_{f,\mathcal{L},g}(\mathbf{x}_i, h(\mathbf{x}_i), \mathbf{x}_j, y_j) &= \mathbf{Tr}\left[\left(\nabla_{y_i'} \mathcal{L}_t \nabla_z g_t\right) \tilde{K}_f(\mathbf{x}_i, \mathbf{x}_j) \left(\nabla_{y_j'} \mathcal{L}_u \nabla_z g_u\right)^\top\right] \\
&= \mathbf{Tr}\left[\tilde{K}_f(\mathbf{x}_i, \mathbf{x}_j) \left(\nabla_{y_j'} \mathcal{L}_u \nabla_z g_u\right)^\top \left(\nabla_{y_i'} \mathcal{L}_t \nabla_z g_t\right)\right] \\
&= \mathbf{Tr}\left[\tilde{K}_f(\mathbf{x}_i, \mathbf{x}_j) \nabla_z^\top g_u \nabla_{y_j'}^\top \mathcal{L}_u \nabla_{y_i'} \mathcal{L}_t \nabla_z g_t\right] \\
&= \mathbf{Tr}\left[\tilde{K}_f(\mathbf{x}_i, \mathbf{x}_j) \tilde{K}_{\mathcal{L},g}^\top(h(\mathbf{x}_j), y_j)\right] \\
&= \sum_{r=1}^{O} \sum_{s=1}^{O} \left[\tilde{K}_f(\mathbf{x}_i, \mathbf{x}_j) \circ \tilde{K}_{\mathcal{L},g}(h(\mathbf{x}_j), y_j)\right]_{rs}
\end{aligned} \tag{17}$$

Therefore, the kernel function $\tilde{k}_{f,\mathcal{L},g}(\mathbf{x}_i, h(\mathbf{x}_i), \mathbf{x}_j, y_j)$ can be viewed as a kernel function implicitly constructed based on the task type and semantics. It consists of two parts, $\tilde{K}_f(\mathbf{x}_i, \mathbf{x}_j)$ and $\tilde{K}_{\mathcal{L},g}(h(\mathbf{x}_j), y_j)$, which correspond to the similarity between the inputs $\mathbf{x}$ of the samples and the importance of different pairs of similarities for solving the task respectively. Specifically,

- $\tilde{K}_f(\mathbf{x}_i, \mathbf{x}_j)$ is a matrix of shape $O \times O$ as

$$
\tilde{K}_f(\mathbf{x}_i, \mathbf{x}_j) = \begin{bmatrix} \langle \nabla_\theta f_1(\mathbf{x}_i), \nabla_\theta f_1(\mathbf{x}_j) \rangle & \cdots & \langle \nabla_\theta f_1(\mathbf{x}_i), \nabla_\theta f_O(\mathbf{x}_j) \rangle \\ \langle \nabla_\theta f_2(\mathbf{x}_i), \nabla_\theta f_1(\mathbf{x}_j) \rangle & \cdots & \langle \nabla_\theta f_2(\mathbf{x}_i), \nabla_\theta f_O(\mathbf{x}_j) \rangle \\ \vdots & \ddots & \vdots \\ \langle \nabla_\theta f_O(\mathbf{x}_i), \nabla_\theta f_1(\mathbf{x}_j) \rangle & \cdots & \langle \nabla_\theta f_O(\mathbf{x}_i), \nabla_\theta f_O(\mathbf{x}_j) \rangle \end{bmatrix}, \quad (18)
$$

  where $f_o(x)$ denotes the $o$-th output of the neural network $f(\mathbf{x}, \theta) \in \mathbb{R}^O$ and $O$ denotes the dimension of the feature space $\tilde{\mathcal{Z}}$. The elements

$$
\tilde{K}_f(\mathbf{x}_i, \mathbf{x}_j)_{rs} = \langle \nabla_\theta f_r(\mathbf{x}_i), \nabla_\theta f_s(\mathbf{x}_j) \rangle, \quad \text{for any } r, s \in [O] \quad (19)
$$

  characterize the similarity between $\mathbf{x}_i$ and $\mathbf{x}_j$ from the perspective of pairs of features, $O^2$ pairs in total, through empirical neural tangent kernel.

- The gradient $\nabla_{y'}\mathcal{L} \nabla_z g$ is a vector such that

$$
\nabla_{y'}\mathcal{L} \nabla_z g = \begin{bmatrix} \sum_{k=1}^{|\mathcal{Y}|} \frac{\partial \mathcal{L}}{\partial g_k} \frac{\partial g_k}{\partial z_1} & \cdots & \sum_{k=1}^{|\mathcal{Y}|} \frac{\partial \mathcal{L}_1}{\partial g_k} \frac{\partial g_k}{\partial z_O} \end{bmatrix}^\top \in \mathbb{R}^{1 \times O}, \quad (20)
$$

  where $z = f(\mathbf{x}_i) \in \mathbb{R}^O$. And the gradient components

$$
(\nabla_{y'}\mathcal{L} \nabla_z g)_o = \sum_{k=1}^{|\mathcal{Y}|} \frac{\partial \mathcal{L}}{\partial g_k} \frac{\partial g_k}{\partial z_o}, \quad \text{for any } o \in [O], \quad (21)
$$

  describe the relationship between the features $z_o$ and solving the task, with some features being more helpful the larger they are and others the opposite.

- Finally, for the matrix $\tilde{K}_{\mathcal{L},g}(h(\mathbf{x}_j), y_j) = (\nabla_{y_i'}\mathcal{L}_t \nabla_z g_t)^\top (\nabla_{y_j'}\mathcal{L}_u \nabla_z g_u)$,

$$
\tilde{K}_{\mathcal{L},g} = \begin{bmatrix} \sum_{k=1}^{|\mathcal{Y}_t|} \frac{\partial \mathcal{L}_t}{\partial g_{t,k}} \frac{\partial g_{t,k}}{\partial z_1} \\ \vdots \\ \sum_{k=1}^{|\mathcal{Y}_t|} \frac{\partial \mathcal{L}_t}{\partial g_{t,k}} \frac{\partial g_{t,k}}{\partial z_O} \end{bmatrix} \begin{bmatrix} \sum_{k=1}^{|\mathcal{Y}_u|} \frac{\partial \mathcal{L}_u}{\partial g_{u,k}} \frac{\partial g_{u,k}}{\partial z_1} & \cdots & \sum_{k=1}^{|\mathcal{Y}_u|} \frac{\partial \mathcal{L}_u}{\partial g_{u,k}} \frac{\partial g_{u,k}}{\partial z_O} \end{bmatrix} \in \mathbb{R}^{O \times O}
$$

$$(22)$$

  encourages similarity between pairs of features sharing the same relationship with solving the task and penalizes the similarity of the opposite.

Therefore, the kernel function simultaneously incorporates input data $\mathbf{x}$, model predictions $h(\mathbf{x})$, and task semantics. The greater the consistency in the input-output relationship between two samples, the higher the kernel function value. For the model vector $\boldsymbol{\tau}_h$ and the task vector $\boldsymbol{\tau}_u$, we have

$$
\langle \boldsymbol{\tau}_h, \boldsymbol{\tau}_u \rangle = \sum_{(\mathbf{x}_i, y_i) \in \mathcal{D}_t} \sum_{(\mathbf{x}_j, y_j) \in \mathcal{D}_u} \tilde{k}_{f,\mathcal{L},g}(\mathbf{x}_i, h(\mathbf{x}_i), \mathbf{x}_j, y_j). \quad (23)
$$

To eliminate the bias introduced by dataset size and optimization trajectory length, we normalize $\boldsymbol{\tau}_h$ and $\boldsymbol{\tau}_u$ to unit vectors. Consequently, $\langle \boldsymbol{\tau}_h, \boldsymbol{\tau}_u \rangle$ precisely corresponds to $\cos(\boldsymbol{\tau}_h, \boldsymbol{\tau}_u)$ in Equation (2).

# B  DETAILS ON THE APPROXIMATION OF PARAMETER VECTOR SIMILARITY

In this section, we establish an error bound for approximating the similarity between parameter vectors, as stated in Theorem 4, and provide a complete proof.

Before delving into the theorems and their proofs, we first define the necessary notation. Consider a pre-trained model, $f(\mathbf{x}, \theta_0 + \boldsymbol{\tau})$ which consists of $L$ weight matrices. Without loss of generality, we assume that each weight matrix has the shape $m \times n$. The corresponding approximated parameter vector $\boldsymbol{\tau}$ is expressed as

$$\boldsymbol{\tau} \approx \tilde{\boldsymbol{\tau}} \triangleq [\Delta\mathbf{W}^1 \, \Delta\mathbf{W}^2 \, \ldots \, \Delta\mathbf{W}^L] = [(\mathbf{B}^1\mathbf{A}^1) \, (\mathbf{B}^2\mathbf{A}^2) \, \ldots \, (\mathbf{B}^L\mathbf{A}^L)] \triangleq \mathbf{B}\mathbf{A},$$

where $\mathbf{A}$ is initialized randomly, and $\mathbf{B}$ is initialized with zeros. To clarify, the dimensions of the relevant matrices are as follows:

$$\Delta\mathbf{W}^l \in \mathbb{R}^{m \times n}, \quad \mathbf{A}^l \in \mathbb{R}^{r \times n}, \quad \mathbf{B}^l \in \mathbb{R}^{m \times r}, \quad \text{for any } l \in [L].$$

Accordingly, the parameter vector $\boldsymbol{\tau} \in \mathbb{R}^{m \times (nL)}$ is the concatenation of matrices $\{\mathbf{W}^l\}$ and $\mathbf{B} \in \mathbb{R}^{m \times (rL)}$ is the concatenation of matrices $\{\mathbf{B}^l\}$. And we obtain the parameter vector $\mathbf{B}$ via

$$\mathbf{B}^* = \arg\min_{\mathbf{B}} \sum_{(\mathbf{x},y) \in \mathcal{D}} \mathcal{L}(g \circ f(\mathbf{x}, \theta_0 + \mathbf{B}\mathbf{A}), h(\mathbf{x})). \tag{24}$$

For simplicity, in this paper, we implicitly flatten matrices from the matrix space to the vector space when computing the inner product and cosine similarity between them. Then we can begin our proof.

**Theorem 4.** *In order to compute the cosine similarity between the parameter vectors $\tilde{\boldsymbol{\tau}} = \mathbf{B}\mathbf{A}$ obtained by the low-rank approximation following Equation (24), the cosine similarity between $\mathbf{B}$ only is sufficient to approximate it efficiently. Specifically, we can prove that for any $\epsilon > 0$,*

$$-2\epsilon + \min\left(\frac{1}{1+\epsilon}\cos(\mathbf{B}_1, \mathbf{B}_2), \frac{1}{1-\epsilon}\cos(\mathbf{B}_1, \mathbf{B}_2)\right) \leq \cos(\tilde{\boldsymbol{\tau}}_1, \tilde{\boldsymbol{\tau}}_2)$$
$$\leq \max\left(\frac{1}{1-\epsilon}\cos(\mathbf{B}_1, \mathbf{B}_2), \frac{1}{1+\epsilon}\cos(\mathbf{B}_1, \mathbf{B}_2)\right) + 2\epsilon, \tag{25}$$

*with probability of $1 - \delta$, when $nL = O(\epsilon^{-2}\ln(\frac{4}{\delta}))$ and $\ln r \leq \frac{\epsilon^2 m}{36\|\mathbf{B}^l\|^2}$ for each $l \in [N]$ and the matrices*

$$\mathbf{A} \sim \mathcal{U}(-\sqrt{3s/n}, +\sqrt{3s/n})$$

*element-wisely with constant factor $s$, which is Kaiming initialization (He et al., 2015) with proper $s$.*

*Proof.* Our proof is divided into three main steps. First, we demonstrate that the random matrix $\mathbf{A}^l(\mathbf{A}^l)^\top$ approximates a scalar matrix $s\mathbf{I}_{r \times r}$ for any $l \in [L]$. Next, we show that the squared norm of $\mathbf{B}^l$, denoted as $\|\mathbf{B}^l\|^2$, is approximately equal to $\|\mathbf{B}^l\mathbf{A}^l\|^2$ for any $l \in [L]$. Finally, we establish that $\cos(\mathbf{B}_1, \mathbf{B}_2)$ closely approximates $\cos(\tilde{\boldsymbol{\tau}}_1, \tilde{\boldsymbol{\tau}}_2)$ with high probability.

**Step 1.** We begin by analyzing the random matrix $\mathbf{A}^l(\mathbf{A}^l)^\top$. It can be decomposed into the sum of independent random matrices as follows:

$$\mathbf{A}^l(\mathbf{A}^l)^\top = \sum_{i \in [n]} a_i a_i^\top \in \mathbb{R}^{r \times r},$$

where $\mathbf{A}^l = [a_1, a_2, \ldots, a_n]$ is element-wise uniformly distributed as $\mathcal{U}(-\sqrt{3s/n}, +\sqrt{3s/n})$. In expectation, the random matrix satisfies:

$$\mathbb{E}[a_i a_i^\top] = \frac{s}{n}\mathbf{I}_{r \times r}, \quad \text{and} \quad s\mathbf{I} \preceq \mathbb{E}[\mathbf{A}^l(\mathbf{A}^l)^\top] \preceq s\mathbf{I}. \tag{26}$$

Moreover, for any vector $v$, it holds that:

$$0 \leq v^\top(a_i a_i^\top)v = (v^\top a_i)^2 \leq \|v\|^2\|a_i\|^2 \leq v^\top\left(\frac{3s}{n}\mathbf{I}\right)v. \tag{27}$$

Thus, the spectrum of $a_i a_i^\top$ is bounded, and we have:

$$\mathbf{0} \preceq a_i a_i^\top \preceq \frac{3s}{n}\mathbf{I}, \quad \text{almost surely.} \tag{28}$$

Next, we apply a tail bound for the sum of random matrices, as stated in Lemma 5 (Tropp, 2012).

**Lemma 5.** *Let* $\mathbf{X}_1, \ldots, \mathbf{X}_n$ *be independent* $r \times r$ *real symmetric matrices with* $0 \preceq \mathbf{X}_i \preceq R \cdot \mathbf{I}$ *for some* $R \in \mathbb{R}$. *Let* $\mu_{\min}\mathbf{I} \preceq \sum_{i=1}^{m} \mathbb{E}[\mathbf{X}_i] \preceq \mu_{\max}\mathbf{I}$. *For any* $0 < \epsilon \leq 1$,

$$\Pr\left(\lambda_{\max}\left(\sum_{i=1}^{n} \mathbf{X}_i\right) \geq (1+\epsilon)\mu_{\max}\right) \leq r \exp\left(-\frac{\epsilon^2 \mu_{\max}}{3R}\right),$$

$$\Pr\left(\lambda_{\min}\left(\sum_{i=1}^{n} \mathbf{X}_i\right) \leq (1-\epsilon)\mu_{\min}\right) \leq r \exp\left(-\frac{\epsilon^2 \mu_{\min}}{2R}\right).$$

By choosing $R = \frac{3s}{n}$ and $\mu_{\max} = \mu_{\min} = s$, we obtain:

$$\Pr\left[(1-\epsilon)s\mathbf{I} \preceq \mathbf{A}^l(\mathbf{A}^l)^\top \preceq (1+\epsilon)s\mathbf{I}\right] \geq 1 - 2r\exp\left(-\frac{\epsilon^2 n}{9s}\right), \tag{29}$$

for any $\epsilon > 0$. This shows that $\mathbf{A}^l(\mathbf{A}^l)^\top$ is approximately equal to $s\mathbf{I}$ with high probability $1 - 2r\exp(-\frac{\epsilon^2 n}{9s})$ for any $l \in [L]$.

**Step 2.** We gonna show that the squared norm of $\mathbf{B}^l$, denoted as $\|\mathbf{B}^l\|^2$, is approximately equal to $\|\mathbf{B}^l\mathbf{A}^l\|^2$ for any $l \in [L]$. Using the decomposition of $\|\mathbf{B}^l\mathbf{A}^l\|^2$, we have:

$$\|\mathbf{B}^l\mathbf{A}^l\|^2 = \langle \mathbf{B}^l\mathbf{A}^l, \mathbf{B}^l\mathbf{A}^l \rangle = \sum_{i\in[n]} b_i^\top (\mathbf{A}^l(\mathbf{A}^l)^\top) b_i. \tag{30}$$

From the previous result, it follows that:

$$(1-\epsilon)\sum_{i\in[n]} b_i^\top b_i \leq \|\mathbf{B}^l\mathbf{A}^l\|^2 \leq (1+\epsilon)\sum_{i\in[n]} b_i^\top b_i, \tag{31}$$

or equivalently we have:

$$(1-\epsilon)\|\mathbf{B}^l\|^2 \leq \|\mathbf{B}^l\mathbf{A}^l\|^2 \leq (1+\epsilon)\|\mathbf{B}^l\|^2, \tag{32}$$

with high probability of $1 - 2r\exp(-\frac{\epsilon^2 m}{9s})$. By moving $r$ into the exp function and assuming that $\ln r \leq \frac{\epsilon^2 m}{36\|\mathbf{B}^l\|^2}$, it follows that

$$\begin{aligned}\Pr[|\|\mathbf{B}^l\mathbf{A}^l\|^2 - \|\mathbf{B}^l\|^2| \geq t] &\leq 2\exp\left(-\frac{t^2 m}{9s\|\mathbf{B}^l\|^4} + \ln r\right) \\ &\leq 2\exp\left(-\frac{t^2 m}{12s\|\mathbf{B}^l\|^4}\right).\end{aligned} \tag{33}$$

It implies that $\|\mathbf{B}^l\mathbf{A}^l\|^2 - \|\mathbf{B}^l\|^2$ is a sub-gaussian random variable with parameter $\sqrt{6s/m}\|\mathbf{B}^l\|$.

**Lemma 6.** *Suppose that the random variables* $\{X_i\}_{i=1}^n$ *are independent and* $\mathbb{E}[X_i] = \mu_i$, *and* $X_i$ *is sub-Gaussian random variable with parameter* $\sigma_i$. *Then for all* $t \geq 0$, *we have*

$$\Pr\left[\left|\sum_{i=1}^n (X_i - \mu_i)\right| \geq t\right] \leq 2\exp\left(\frac{-t^2}{2\sum_{i=1}^n \sigma_i^2}\right).$$

With Lemma 6 (Mohri et al., 2012), the sum of sub-gaussian random variables $\|\mathbf{B}_l\mathbf{A}_l\|^2$ in all $N$ layers $\|\mathbf{BA}\|^2$ satisfies

$$\Pr[|\|\mathbf{BA}\|^2 - \|\mathbf{B}\|^2| \geq t] \leq 2\exp\left(-\frac{t^2 m}{12s\sum_{l=1}^L \|\mathbf{B}_l\|^4}\right) \tag{34}$$

In general, for n real numbers $x_1, \ldots, x_n$, we have the inequality:

$$\frac{1}{n}(x_1 + x_2 + \cdots + x_n)^2 \leq x_1^2 + x_2^2 + \cdots + x_n^2 \leq (x_1 + x_2 + \cdots + x_n)^2 \tag{35}$$

where the first inequality takes equality when $x_i = \frac{1}{n}\sum_i^n x_i$, and the second inequality takes equality when some $x_i = \sum_i^n x_i$. In practice, we find that $\|\mathbf{B}\|^2$ is nearly uniformly contributed by $\|\mathbf{B}^l\|^2$ in each layer. Therefore, we can assume that there exists a constant $c \ll L$ such that

$$\|\mathbf{B}^1\|^4 + \|\mathbf{B}^2\|^4 + \cdots + \|\mathbf{B}^L\|^4 \le \frac{c}{L}(\|\mathbf{B}^1\|^2 + \|\mathbf{B}^2\|^2 + \cdots + \|\mathbf{B}^L\|^2)^2 = \frac{c}{L}\|\mathbf{B}\|^4 \quad (36)$$

Finally, summing across all layers, the L2 norm of the task vector satisfies:

$$(1-\epsilon)\|\mathbf{B}\|^2 \le \|\mathbf{B}\mathbf{A}\|^2 \le (1+\epsilon)\|\mathbf{B}\|^2, \quad (37)$$

with high probability of $1 - 2e^{\epsilon^2 nL/12sc}$.

**Step 3.** We will establish that $\cos(\mathbf{B}_1, \mathbf{B}_2)$ closely approximates $\cos(\tilde{\tau}_1, \tilde{\tau}_2)$ with high probability. First, by substituting $\mathbf{B} = \mathbf{B}_1 - \mathbf{B}_2$ and $\mathbf{B} = \mathbf{B}_1 + \mathbf{B}_2$ into Equation (37), we obtain the following inequalities:

$$\begin{aligned}(1-\epsilon)\|\mathbf{B}_1 - \mathbf{B}_2\|^2 \le \|\mathbf{B}_1\mathbf{A} - \mathbf{B}_2\mathbf{A}\|^2 \le (1+\epsilon)\|\mathbf{B}_1 - \mathbf{B}_2\|^2, \\ (1-\epsilon)\|\mathbf{B}_1 + \mathbf{B}_2\|^2 \le \|\mathbf{B}_1\mathbf{A} + \mathbf{B}_2\mathbf{A}\|^2 \le (1+\epsilon)\|\mathbf{B}_1 + \mathbf{B}_2\|^2.\end{aligned} \quad (38)$$

Next, to bound the inner product, note that

$$\langle \mathbf{B}_1\mathbf{A}, \mathbf{B}_2\mathbf{A}\rangle = \frac{1}{4}\left(\|\mathbf{B}_1\mathbf{A} + \mathbf{B}_2\mathbf{A}\|^2 - \|\mathbf{B}_1\mathbf{A} - \mathbf{B}_2\mathbf{A}\|^2\right) \le \langle \mathbf{B}_1, \mathbf{B}_2\rangle + \frac{\epsilon}{2}\left(\|\mathbf{B}_1\|^2 + \|\mathbf{B}_2\|^2\right), \quad (39)$$

and for the other direction,

$$\langle \mathbf{B}_1\mathbf{A}, \mathbf{B}_2\mathbf{A}\rangle = \frac{1}{4}\left(\|\mathbf{B}_1\mathbf{A} + \mathbf{B}_2\mathbf{A}\|^2 - \|\mathbf{B}_1\mathbf{A} - \mathbf{B}_2\mathbf{A}\|^2\right) \ge \langle \mathbf{B}_1, \mathbf{B}_2\rangle - \frac{\epsilon}{2}\left(\|\mathbf{B}_1\|^2 + \|\mathbf{B}_2\|^2\right). \quad (40)$$

Now, consider the case $\langle \mathbf{B}_1\mathbf{A}, \mathbf{B}_2\mathbf{A}\rangle > 0$. To bound the cosine similarity, we have

$$\frac{\langle \mathbf{B}_1\mathbf{A}, \mathbf{B}_2\mathbf{A}\rangle}{\|\mathbf{B}_1\mathbf{A}\|\|\mathbf{B}_2\mathbf{A}\|} \le \frac{\langle \mathbf{B}_1, \mathbf{B}_2\rangle}{(1-\epsilon)\|\mathbf{B}_1\|\|\mathbf{B}_2\|} + \frac{\epsilon}{2}\frac{\|\mathbf{B}_1\|^2 + \|\mathbf{B}_2\|^2}{(1-\epsilon)\|\mathbf{B}_1\|\|\mathbf{B}_2\|}, \quad (41)$$

and similarly,

$$\frac{\langle \mathbf{B}_1\mathbf{A}, \mathbf{B}_2\mathbf{A}\rangle}{\|\mathbf{B}_1\mathbf{A}\|\|\mathbf{B}_2\mathbf{A}\|} \ge \frac{\langle \mathbf{B}_1, \mathbf{B}_2\rangle}{(1+\epsilon)\|\mathbf{B}_1\|\|\mathbf{B}_2\|} - \frac{\epsilon}{2}\frac{\|\mathbf{B}_1\|^2 + \|\mathbf{B}_2\|^2}{(1+\epsilon)\|\mathbf{B}_1\|\|\mathbf{B}_2\|}. \quad (42)$$

For the case $\langle \mathbf{B}_1\mathbf{A}, \mathbf{B}_2\mathbf{A}\rangle < 0$, we establish similar bounds:

$$\frac{\langle \mathbf{B}_1\mathbf{A}, \mathbf{B}_2\mathbf{A}\rangle}{\|\mathbf{B}_1\mathbf{A}\|\|\mathbf{B}_2\mathbf{A}\|} \le \frac{\langle \mathbf{B}_1, \mathbf{B}_2\rangle}{(1+\epsilon)\|\mathbf{B}_1\|\|\mathbf{B}_2\|} + \frac{\epsilon}{2}\frac{\|\mathbf{B}_1\|^2 + \|\mathbf{B}_2\|^2}{(1+\epsilon)\|\mathbf{B}_1\|\|\mathbf{B}_2\|}, \quad (43)$$

and,

$$\frac{\langle \mathbf{B}_1\mathbf{A}, \mathbf{B}_2\mathbf{A}\rangle}{\|\mathbf{B}_1\mathbf{A}\|\|\mathbf{B}_2\mathbf{A}\|} \ge \frac{\langle \mathbf{B}_1, \mathbf{B}_2\rangle}{(1-\epsilon)\|\mathbf{B}_1\|\|\mathbf{B}_2\|} - \frac{\epsilon}{2}\frac{\|\mathbf{B}_1\|^2 + \|\mathbf{B}_2\|^2}{(1-\epsilon)\|\mathbf{B}_1\|\|\mathbf{B}_2\|}. \quad (44)$$

By combining these bounds, and letting $\Delta(\|\mathbf{B}_1\|, \|\mathbf{B}_2\|) = \frac{\|\mathbf{B}_1\|}{\|\mathbf{B}_2\|} + \frac{\|\mathbf{B}_2\|}{\|\mathbf{B}_1\|}$, we derive the following upper bound:

$$\begin{aligned}\cos(\mathbf{B}_1\mathbf{A}, \mathbf{B}_2\mathbf{A}) \le{}& \max\left(\frac{1}{1-\epsilon}\cos(\mathbf{B}_1, \mathbf{B}_2), w\frac{1}{1+\epsilon}\cos(\mathbf{B}_1, \mathbf{B}_2)\right) \\ &+ \max\left(\frac{\epsilon}{2(1-\epsilon)}\Delta(\|\mathbf{B}_1\|, \|\mathbf{B}_2\|), \frac{\epsilon}{2(1+\epsilon)}\Delta(\|\mathbf{B}_1\|, \|\mathbf{B}_2\|)\right) \\ \le{}& \max\left(\frac{1}{1-\epsilon}\cos(\mathbf{B}_1, \mathbf{B}_2), \frac{1}{1+\epsilon}\cos(\mathbf{B}_1, \mathbf{B}_2)\right) + \frac{\epsilon}{2(1-\epsilon)}\Delta(\|\mathbf{B}_1\|, \|\mathbf{B}_2\|).\end{aligned} \quad (45)$$

The lower bound can be proved in a similar manner.

In summary, we establish that

$$
\begin{aligned}
&\min\left(\frac{1}{1+\epsilon}\cos(\mathbf{B}_1,\mathbf{B}_2),\ \frac{1}{1-\epsilon}\cos(\mathbf{B}_1,\mathbf{B}_2)\right) - \frac{\epsilon}{2(1+\epsilon)}\Delta(\|\mathbf{B}_1\|,\|\mathbf{B}_2\|)\\
&\leq \cos(\mathbf{B}_1\mathbf{A},\mathbf{B}_2\mathbf{A})\\
&\leq \max\left(\frac{1}{1-\epsilon}\cos(\mathbf{B}_1,\mathbf{B}_2),\ \frac{1}{1+\epsilon}\cos(\mathbf{B}_1,\mathbf{B}_2)\right) + \frac{\epsilon}{2(1-\epsilon)}\Delta(\|\mathbf{B}_1\|,\|\mathbf{B}_2\|).
\end{aligned}
\tag{46}
$$

Since we only care about the direction of the parameter vector, without loss of generality we let $\|\mathbf{B}_1\| = \|\mathbf{B}_2\| = 1$ and $\epsilon \leq 0.5$, the result simplifies to

$$
\begin{aligned}
-2\epsilon + \min\left(\frac{1}{1+\epsilon}\cos(\mathbf{B}_1,\mathbf{B}_2),\ \frac{1}{1-\epsilon}\cos(\mathbf{B}_1,\mathbf{B}_2)\right) &\leq \cos(\mathbf{B}_1\mathbf{A},\mathbf{B}_2\mathbf{A})\\
&\leq \max\left(\frac{1}{1-\epsilon}\cos(\mathbf{B}_1,\mathbf{B}_2),\ \frac{1}{1+\epsilon}\cos(\mathbf{B}_1,\mathbf{B}_2)\right) + 2\epsilon,
\end{aligned}
\tag{47}
$$

with high probability of $1 - 4e^{\epsilon^2 nL/12sc}$. $\qquad\square$

## C    EXPERIMENTAL SETTINGS

### C.1    DATASET DESCRIPTIONS

**Classic NLP.** The classic natural language processing datasets we use are mainly from the GLUE benchmark datasets. Specific information about these datasets is listed below:

- **CoLA** (Corpus of Linguistic Acceptability): A dataset for grammatical acceptability classification, where each English sentence is labeled as either grammatically correct or incorrect, sourced from linguistic literature.

- **MNLI** (Multi-Genre Natural Language Inference): A natural language inference dataset where models determine if a hypothesis entails, contradicts, or is neutral with respect to a given premise. It includes diverse genres of text.

- **MRPC** (Microsoft Research Paraphrase Corpus): A dataset for paraphrase identification, containing sentence pairs annotated for semantic equivalence, useful for measuring sentence similarity.

- **QNLI** (Question Natural Language Inference): A dataset derived from SQuAD, reformulated as a binary classification task to determine if a sentence contains the answer to a given question.

- **QQP** (Quora Question Pairs): A paraphrase detection dataset consisting of question pairs from Quora, labeled as either duplicate (same meaning) or not.

- **RTE** (Recognizing Textual Entailment): A natural language inference dataset where models classify whether a given premise entails a hypothesis, sourced from various textual entailment challenges.

- **SNLI** (Stanford Natural Language Inference Corpus): A large-scale dataset for natural language inference, with sentence pairs labeled as entailment, contradiction, or neutral.

- **SST-2** (Stanford Sentiment Treebank - Binary Classification): A sentiment analysis dataset where sentences from movie reviews are labeled as either positive or negative.

- **STS-B** (Semantic Textual Similarity Benchmark): A dataset for evaluating the semantic similarity between sentence pairs, with similarity scores ranging from 0 to 5.

- **WNLI** (Winograd Natural Language Inference): A challenging dataset derived from the Winograd Schema Challenge, requiring commonsense reasoning to resolve ambiguous pronoun references.

We also evaluate our method on some other natural language processing datasets:

- **cr** (Customer Reviews Dataset): A sentiment analysis dataset containing customer reviews from various products, labeled as positive or negative.

- **mpqa** (Multi-Perspective Question Answering Opinion Corpus): A dataset focused on opinion mining, containing news articles annotated for subjective expressions and sentiment.

- **mr** (Movie Reviews Dataset): A dataset for sentiment analysis, where movie reviews are labeled as either positive or negative.

- **subj** (Subjectivity Dataset): A dataset for classifying sentences as subjective (expressing opinions) or objective (stating factual information).

- **trec** (Text REtrieval Conference Question Classification Dataset): A dataset for question classification, where questions are categorized into different types such as location, person, number, and more.

**Computer vision.** We evaluated our method on several computer vision datasets. These datasets are presented as follows

- **CIFAR-10**: A widely used image classification dataset containing 60,000 small (32×32) color images across 10 categories, including airplanes, cars, and animals.

- **RESISC45** (Remote Sensing Image Scene Classification): A dataset with 31,500 high-resolution aerial images categorized into 45 scene types (e.g., airport, forest, residential areas), designed for remote sensing applications.

- **MNIST** (Modified National Institute of Standards and Technology Database): A classic dataset of 70,000 grayscale handwritten digits (0-9) in 28×28 resolution, commonly used for benchmarking digit classification models.

- **SUN397** (Scene UNderstanding Dataset): A large dataset with 108,754 images covering 397 different scene categories, used for scene recognition and understanding tasks.

- **SVHN** (Street View House Numbers): A dataset containing over 600,000 real-world images of house numbers from Google Street View, designed for digit recognition in natural scenes.

- **GTSRB** (German Traffic Sign Recognition Benchmark): A dataset of over 50,000 images containing 43 different German traffic sign categories, used for traffic sign classification tasks. Food-101: A large dataset containing 101,000 images across 101 different food categories, such as sushi, pizza, and spaghetti, used for food recognition.

- **EuroSAT**: A satellite image dataset with 27,000 RGB images categorized into 10 land use classes, such as residential areas, forests, and industrial zones, based on Sentinel-2 satellite data.

- **DTD** (Describable Textures Dataset): A dataset containing 5,640 texture images categorized into 47 different texture descriptions, such as "cracked" or "striped", used for material recognition.

- **Stanford Cars** (Cars Dataset): A dataset with 16,185 images of 196 different car models, categorized by make, model, and year, used for fine-grained vehicle classification.

- **CIFAR-100**: An extension of CIFAR-10, containing 60,000 images classified into 100 fine-grained categories, grouped into 20 superclasses (e.g., mammals, vehicles, insects).

- **STL-10**: A dataset similar to CIFAR-10 but with larger (96×96) images and fewer labeled examples, designed for semi-supervised learning tasks.

**Medical LLM.** We construct learnwares based on the following supervised fine-tuning (STF) datasets for solving the medical LLM benchmarks:

- **AlpaCare**: The AlpaCare-MedInstruct-52k dataset, available under lavita/AlpaCare-MedInstruct-52k, consists of 52,000 instruction-based medical dialogues. It is designed to improve medical language understanding and response generation, focusing on clinical inquiries and patient care instructions.

- **ChatDoctor**: The ChatDoctor-HealthCareMagic-100k dataset, referenced as lavita/ChatDoctor-HealthCareMagic-100k, contains 100,000 medical dialogue interactions. The dataset is collected from HealthCareMagic, providing extensive doctor-patient conversational data for training medical conversational AI models.

- **medalpaca_cleaned**: This dataset combines multiple medical text sources, including: This dataset combines multiple medical text sources, including:

  - *medical_meadow_wikidoc*: Medical encyclopedic knowledge from WikiDoc.
  - *medical_meadow_medical_flashcards*: Structured medical knowledge in flashcard format.
  - *medical_meadow_wikidoc_patient_information*: Patient-oriented medical content extracted from WikiDoc.
  - *medical_meadow_pubmed_causal*: Causal medical information extracted from PubMed.
  - *medical_meadow_mediqa*: Medical question-answering dataset designed for clinical inquiries.
  - *medical_meadow_health_advice*: General medical guidance and advice for patient care.

- **medical_flashcards** The *medical_meadow_medical_flashcards* dataset provides structured medical knowledge in a flashcard format, aiding in medical education and knowledge retention.

- **medqa_train** The *medical_meadow_medqa* dataset focuses on multiple-choice question answering in the medical domain. It contains curated medical questions designed to assess and enhance medical reasoning capabilities.

- **pubmed_causal** The *medical_meadow_pubmed_causal* dataset is derived from PubMed, focusing on causal relationships in medical research literature. It is useful for developing models that require causal inference in healthcare.

- **medmcqa_train** The *medmcqa_instruct* dataset (`chenhaodev/medmcqa_instruct`) is a large-scale medical question-answering dataset. It contains multiple-choice questions covering various medical subjects, aiding in AI-driven medical education and assessment.

- **medqa_train & pubmed_causal** This dataset combination includes both *medical_meadow_medqa* and *medical_meadow_pubmed_causal*, integrating medical multiple-choice question answering with causal inference from PubMed literature.

- **AlpaCare & ChatDoctor** This dataset set includes:
  - *LinhDuong/chatdoctor-5k*: A small-scale medical conversational dataset.
  - *ChatDoctor-HealthCareMagic-100k*: Large-scale medical dialogue dataset.
  - *AlpaCare-MedInstruct-52k*: Instruction-based medical response dataset.

- **medalpaca_cleaned & AlpaCare & ChatDoctor** This dataset aggregation merges *medalpaca_cleaned*, *AlpaCare-MedInstruct-52k*, and *ChatDoctor-HealthCareMagic-100k*, offering a comprehensive training set

These evaluation datasets are from the Open Medical-LLM Leaderboard(Pal et al., 2024), and they have been used to evaluate the performance of our method for identifying large language models.

- **MedMCQA**: A large-scale Multiple-Choice Question Answering dataset designed to address real-world medical entrance exam questions. It comprises over 194,000 high-quality multiple-choice questions covering 2,400 healthcare topics and 21 medical subjects, collected from AIIMS and NEET PG entrance exams. Each sample includes a question, correct answer(s), and other options, requiring deep language understanding and reasoning across a wide range of medical subjects and topics.

- **MedQA 4-Options**: A benchmark dataset derived from medical licensing exam questions, formatted as multiple-choice questions with four answer options. It is used to evaluate the performance of language models in understanding and answering medical questions accurately.

- **Anatomy**: A subset of the MMLU (Massive Multitask Language Understanding) benchmark focusing on anatomical knowledge. It assesses a model's understanding of human anatomy through various question formats, including multiple-choice questions.

- **Clinical Knowledge**: Part of the MMLU benchmark, this subset evaluates a model's grasp of clinical concepts and practices. It includes questions that test understanding of clinical scenarios, diagnoses, and treatments.

- **College Biology**: Another subset of the MMLU benchmark, focusing on biology at the college level. It assesses knowledge in areas such as genetics, ecology, and cellular biology through various question types.

- **College Medicine**: A component of the MMLU benchmark that tests knowledge in medical subjects typically covered in college-level courses. It includes questions on topics like pharmacology, pathology, and physiology.

- **Medical Genetics**: This subset of the MMLU benchmark focuses on genetics within the medical field. It evaluates understanding of genetic principles, disorders, and their clinical implications.

- **Professional Medicine**: Part of the MMLU benchmark, this subset assesses advanced medical knowledge expected of practicing professionals. It includes questions on complex medical cases, treatments, and ethical considerations.

- **PubMedQA**: A question-answering dataset collected from PubMed abstracts, focusing on research articles. It contains questions, corresponding abstracts, and answers, designed to evaluate a model's ability to comprehend and extract information from biomedical literature.

### C.2 COMPARED METHODS DETAILS

To demonstrate the generality and effectiveness of our proposed PAVE specification, we selected representative pre-trained models from the corresponding domains for comparison.

**Pre-trained language models.** We compare these following pre-trained models for solving natural language processing tasks. Their specific information is listed below:

- **BERT** (Devlin et al., 2019) is a transformer-based language model developed by Google, designed to capture bidirectional context by processing text from both left to right and right to left simultaneously. It is pre-trained using masked language modeling (MLM) and next sentence prediction (NSP), allowing it to understand word relationships in context. BERT comes in multiple sizes, including BERT-base (12 layers, 768 hidden units, 12 attention heads) and BERT-large (24 layers, 1024 hidden units, 16 attention heads), providing different levels of computational efficiency and model complexity.

- **RoBERTa** (Liu et al., 2019), developed by Facebook AI, is an optimized variant of BERT that removes the next sentence prediction task and trains with larger batch sizes and more data. It refines the masked language modeling objective by dynamically changing the masking pattern across training epochs, leading to improved contextual representations. Like BERT, RoBERTa is available in different sizes, including RoBERTa-base (same architecture as BERT-base) and RoBERTa-large (same architecture as BERT-large), offering enhanced performance in various NLP tasks due to its more extensive pretraining regimen.

**Pre-trained vision models.** These are some common pre-trained visual models for solving computer vision tasks as our compared methods. Their specific information is presented as follows:

- **CLIP** (Radford et al., 2021) is a powerful multimodal model introduced by OpenAI, trained using contrastive learning to align images and text in a shared embedding space. Unlike traditional supervised learning models that rely on labeled datasets, CLIP is trained on large-scale internet data, allowing it to generalize across a wide range of visual tasks. It supports different backbone architectures, including Vision Transformers (ViT), which are defined by patch sizes such as ViT-B/16 and ViT-B/32—where the number refers to the patch size used for processing images. A smaller patch size (e.g., 16) allows for finer granularity but requires more computation, while a larger patch size (e.g., 32) reduces computational cost at the expense of detail. CLIP has demonstrated strong zero-shot learning capabilities, making it highly adaptable across various domains.

- **ResNet** (He et al., 2016), short for Residual Network, is a deep convolutional neural network architecture designed to address the vanishing gradient problem in very deep networks. By introducing residual connections, ResNet allows gradients to bypass multiple layers, ensuring stable training even with hundreds of layers. The architecture comes in various depths, including ResNet-50, ResNet-101, and ResNet-152, where the number indicates the total number of layers in the model. Deeper variants, such as ResNet-152, tend to capture more complex features but require more computational power. ResNet has been a foundational model in computer vision, excelling in image classification, object detection, and feature extraction across various datasets, including ImageNet.

**Large language models.** We also compare our method with some popular large language models on medical LLM benchmarks. These large language models are described as follows:

- Developed by OpenAI, **ChatGPT-3.5 Turbo** (OpenAI, 2023) is an optimized version of GPT-3.5, designed to balance efficiency, speed, and response quality. It is trained using a mixture of supervised fine-tuning and reinforcement learning from human feedback (RLHF), making it adept at conversational AI, coding, and general problem-solving. Compared to its predecessors, it is more cost-effective and faster, often deployed in applications requiring reliable and coherent text generation.

- Meta's **Llama-3-Instruct** (Grattafiori et al., 2024) is a fine-tuned variant of the Llama-3 series, optimized for instruction-following tasks. It builds on Llama-3's transformer-based architecture, with improvements in context understanding and reasoning capabilities. This model is designed to be highly efficient, leveraging a compact architecture to perform well

in knowledge-intensive tasks while maintaining lower computational costs. It is particularly useful for chatbots, research, and enterprise AI applications.

- An iterative improvement over Llama-3, **Llama-3.1-Instruct** (Grattafiori et al., 2024) refines instruction-following accuracy and response relevance. It enhances fine-tuning techniques to provide better alignment with user queries, ensuring more precise and context-aware outputs. Compared to its predecessor, it demonstrates improved robustness in handling complex queries while maintaining high efficiency. This model is suitable for scenarios requiring nuanced responses, such as content generation, tutoring, and technical assistance.

We also compare some learnware identification methods to illustrate the advantages of the PAVE specification. Next, we will introduce these methods and the reasons for comparing with them.

- **Random.** The method uniformly and randomly picks one of the candidate learnwares and return to the user. The performance of method reflects the average capability of the candidate learnwares and can be considered as a baseline.

- **Oracle.** The method always returns to the user the best performing learnware on their user task, and is a method that exists ideally. It reflects the upper limit of what a learnware identification method can achieve and the optimal capability of the candidate learnwares.

- **RKME.** This is the most dominant approach in previous research on *Learnware* and has had a profound impact in the field(Wu et al., 2023; Tan et al., 2024c). The implementation of the method is based on the released platform Beimingwu (Tan et al., 2024b).

## C.3 EXPERIMENT CONFIGURATION

Following the original dataset split, we divide the data into a training set and a test set, merging the validation set into the training set if it exists. The original test set remains unchanged, while the training set is further split into two parts: one for users and one for developers. For user tasks, we adopt the k-shot setting from few-shot learning, where each class contains $k$ samples for the classification task. The regression task contains the same number of samples as the binary classification task. The remaining training samples are allocated to developers for model training and learnware construction. For natural language processing datasets, $k$ is set to 16, while for computer vision datasets, it is set to 2, as they typically contain more classes.

In this experiment, we conducted two separate stages of training: one for learnware models and another for task vector construction, as outlined below.

**For learnware training,** all models were trained using the Adam optimizer (Kingma and Ba, 2015) and followed the hyperparameters listed in Table A3. Specifically, the learning rate was set to 2e-5 for natural language processing tasks and 1e-5 for computer vision tasks, with a weight decay of 0.1 for all tasks. The batch size was set to 8 for NLP tasks and 128 for computer vision tasks. The maximum number of epochs was set to 5 for computer vision tasks, while the maximum number of steps was capped at 1000 for some tasks. In cases where models identified from user tasks were fine-tuned, the same set of training hyperparameters was used to ensure fairness and generalizability across all tasks. This consistent approach in fine-tuning ensures that the models adapted to specific user tasks without introducing variability due to differing training configurations.

**For task vector training,** the hyperparameters largely aligned with those used for learnware training to maintain consistency in how task vectors represent the models' capabilities. The learning rate for task vector training was set to 2e-5 for natural language processing tasks and 1e-5 for computer vision tasks, matching the learning rates used in learnware training. Similarly, the batch size was set to 8 for NLP tasks and 128 for computer vision tasks, consistent with the batch sizes used during learnware training. Unlike learnware training, we did not apply weight decay for NLP and computer vision tasks in task vector training, which aligns with the explanation provided in Section 3.2. Task vectors were constructed using pre-trained models appropriate for each task type. ReBERTa-base was used for encoder-only natural language processing tasks, CLIP-B-32 for computer vision tasks, and qwen-0.5B (Bai et al., 2023) for decoder-only medical LLMs.

Table A3: Hyper-parameters for training learnwares.

| Hyper-parameter | Classic NLP | Computer vision | Medical LLM |
|---|---|---|---|
| Learning rate | 2e-5 | 1e-5 | {1e-5, 2e-5, 1e-4, 2e-4} |
| Weight decay | 0.1 | 0.1 | 0.0 |
| Batch size | 8 | 128 | {32, 64} |
| Number of epochs (max) | - | 5 | 5 |
| Number of steps (max) | 1000 | - | - |
| Base model | RoBERTa-base | ViT-B-32 | Llama-3.1-8B-Instruct |

## C.4 COMPUTATION RESOURCES

Experiments were conducted using a Tesla A100 80GB GPU, two Intel Xeon Platinum 8358 CPUs with 32 cores each (base clock 2.6 GHz, turbo boost 3.4 GHz), and 512 GB of RAM.

# D EXPERIMENTAL RESULTS

This appendix provides additional details and supplementary materials to complement the experimental results presented in Section 4. It includes extended performance metrics, ablation studies, and additional visualizations that further demonstrate the effectiveness of our method across diverse task types. The results are structured to highlight the performance under various experimental settings.

## D.1 PERFORMANCES

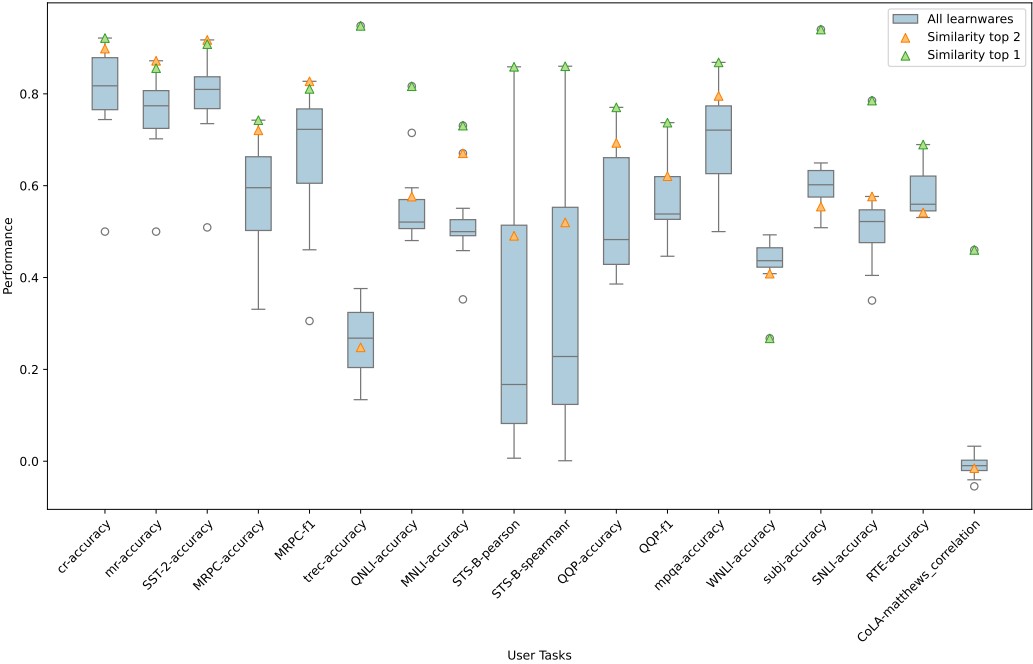

Figure A4: Visualization of performance on classic NLP datasets.

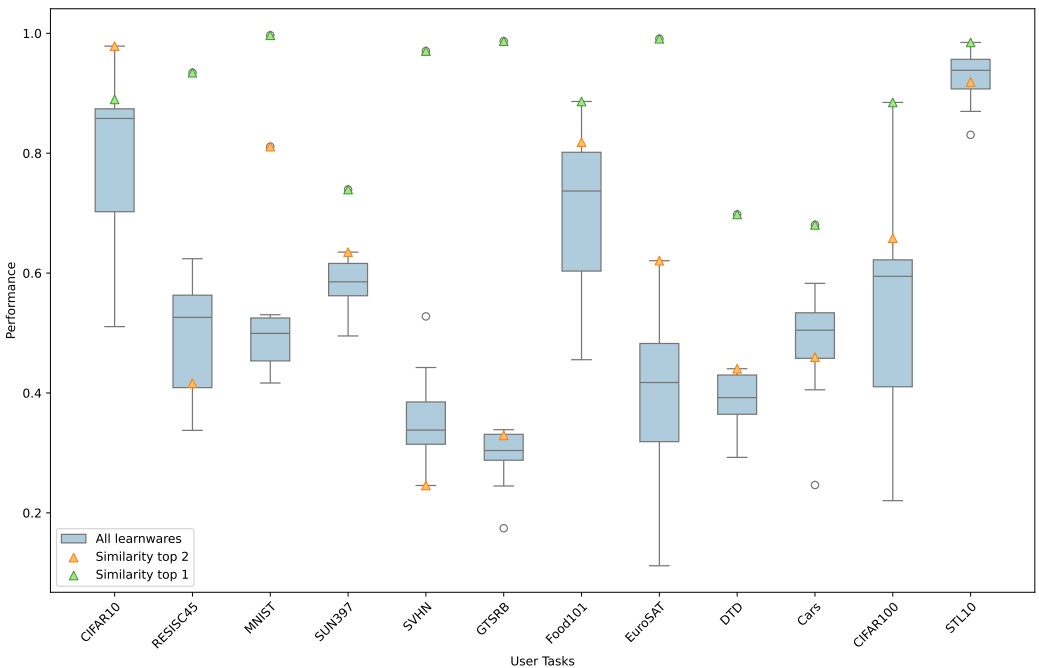

Figure A5: Visualization of performance on computer vision datasets.

Table A4: Performances on medical LLM benchamrks.

| User Task | Large Language Models | | | Identify Learnware | | | Oracle |
|---|---|---|---|---|---|---|---|
| Dataset | GPT-3.5-Turbo | Llama-3-Instruct | Llama-3.1-Instruct | Random | RKME | PAVE | Oracle |
| Anatomy | 65.92 | 60.41 | 68.15 | 68.03 | 68.89 | **70.37** | 70.37 |
| Clinical knowledge | 74.71 | 60.41 | **78.87** | 76.78 | 77.36 | 78.49 | 79.09 |
| College biology | 72.91 | 63.70 | 81.25 | 80.88 | 79.86 | **84.03** | 85.28 |
| College medicine | 64.73 | **71.70** | 68.79 | 66.85 | 69.07 | 67.92 | 70.23 |
| Medical genetics | 74.00 | 74.31 | 77.00 | **78.68** | 76.50 | 78.00 | 82.33 |
| MedMCQA | 53.79 | 57.08 | 59.26 | 58.03 | 56.25 | **60.55** | 60.55 |
| MedQA | 57.71 | 57.08 | 63.63 | 60.94 | **64.24** | 64.13 | 64.24 |
| Professional medicine | 72.79 | 63.01 | **77.57** | 75.93 | 75.37 | 75.07 | 78.68 |
| PubMedQA | 72.66 | **82.00** | 74.80 | 74.59 | 76.20 | 76.50 | 76.50 |
| Avg.(↑) | 67.69 | 65.52 | 72.15 | 71.19 | 71.53 | **72.78** | 74.14 |
| Avg. rank(↓) | 5.44 | 4.56 | 2.44 | 3.44 | 3.00 | **2.11** | - |
| PAVE (win/tie/loss) | 9/0/0 | 7/0/2 | 6/0/3 | 7/0/2 | 6/0/3 | - | 0/3/6 |

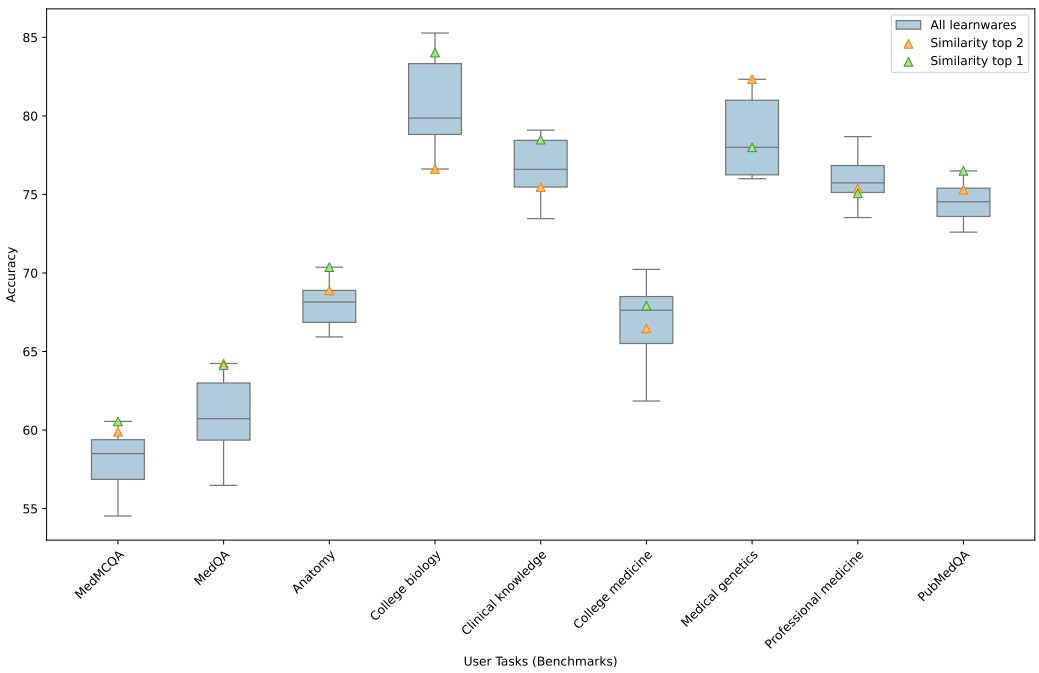

Figure A6: Visualization of performance on medical LLM benchmarks.

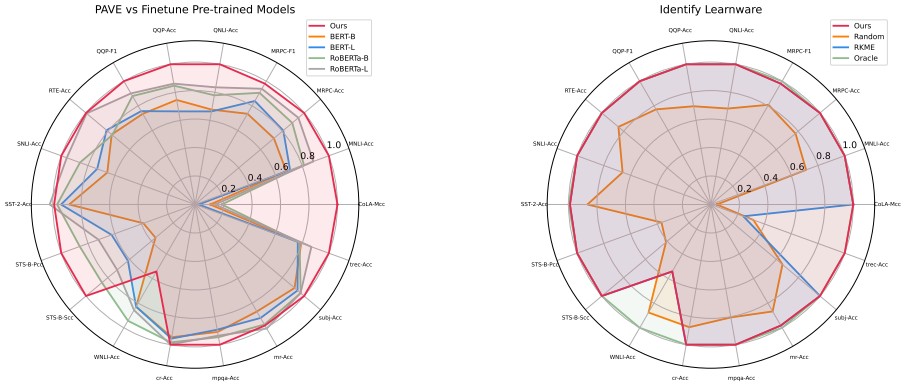

Figure A7: **Performances on classic NLP datasets.** And to achieve better visualization, all performance metrics are normalized relative to the Oracle.

Table A5: Performances on classic NLP datasets.

| User Task | | Finetune Pre-trained Models | | | | Identify Learnware | | | |
| Dataset | Metric | BERT-B | BERT-L | RoBERTa-B | RoBERTa-L | Random | RKME | PAVE | Oracle |
|---|---|---|---|---|---|---|---|---|---|
| CoLA | Mcc | 0.049 | 0.012 | 0.086 | 0.064 | 0.019 | **0.460** | **0.460** | 0.460 |
| MNLI | Acc | 0.498 | 0.519 | 0.594 | 0.646 | 0.520 | **0.731** | **0.731** | 0.731 |
| MRPC | Acc | 0.537 | 0.600 | 0.662 | 0.703 | 0.576 | **0.743** | **0.743** | 0.743 |
| MRPC | F1 | 0.609 | 0.694 | 0.749 | 0.776 | 0.669 | **0.811** | **0.811** | 0.827 |
| QNLI | Acc | 0.552 | 0.542 | 0.636 | 0.682 | 0.559 | **0.817** | **0.817** | 0.817 |
| QQP | Acc | 0.574 | 0.512 | 0.653 | 0.664 | 0.540 | **0.771** | **0.771** | 0.771 |
| QQP | F1 | 0.542 | 0.558 | 0.648 | 0.664 | 0.569 | **0.737** | **0.737** | 0.737 |
| RTE | Acc | 0.527 | 0.560 | 0.523 | 0.686 | 0.585 | **0.690** | **0.690** | 0.690 |
| SNLI | Acc | 0.515 | 0.575 | 0.676 | 0.753 | 0.519 | **0.785** | **0.785** | 0.785 |
| SST-2 | Acc | 0.811 | 0.864 | 0.889 | **0.937** | 0.796 | 0.908 | 0.908 | 0.917 |
| STS-B | Pcc | 0.332 | 0.536 | 0.740 | 0.620 | 0.317 | **0.859** | **0.859** | 0.859 |
| STS-B | Scc | 0.312 | 0.529 | 0.734 | 0.615 | 0.354 | **0.860** | **0.860** | 0.860 |
| WNLI | Acc | 0.408 | 0.408 | **0.465** | 0.423 | 0.432 | 0.268 | 0.268 | 0.493 |
| cr | Acc | 0.873 | 0.881 | 0.906 | 0.920 | 0.806 | **0.921** | **0.921** | 0.921 |
| mpqa | Acc | 0.789 | 0.775 | 0.822 | 0.812 | 0.700 | **0.869** | **0.869** | 0.869 |
| mr | Acc | 0.757 | 0.802 | 0.849 | **0.877** | 0.756 | 0.856 | 0.856 | 0.872 |
| subj | Acc | 0.858 | 0.883 | 0.910 | 0.909 | 0.617 | **0.940** | **0.940** | 0.940 |
| trec | Acc | 0.752 | 0.728 | 0.740 | 0.824 | 0.301 | 0.230 | **0.948** | 0.948 |
| Avg.(↑) | | 0.572 | 0.610 | 0.682 | 0.699 | 0.535 | 0.736 | **0.776** | 0.791 |
| Avg. rank(↓) | | 5.889 | 5.444 | 3.722 | 3.000 | 5.833 | 1.722 | **1.389** | - |
| PAVE (win/tie/loss) | | 17/0/1 | 17/0/1 | 17/0/1 | 15/0/3 | 17/0/1 | 1/17/0 | - | 0/14/4 |

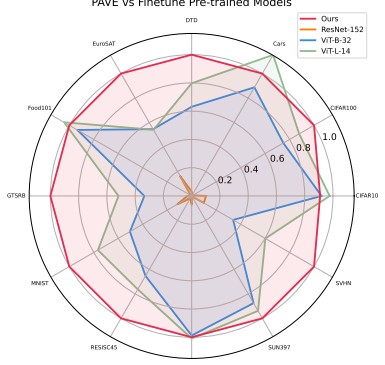 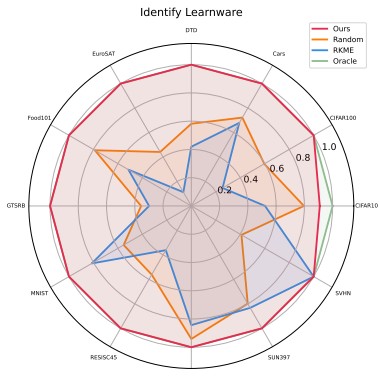

Figure A8: **Performances on computer vision datasets.** And to achieve better visualization, all performance metrics are normalized relative to the Oracle.

Table A6: Performances on computer vision datasets.

| User Task | Finetune Pre-trained Models | | | Identify Learnware | | | |
| Dataset | ResNet-152 | ViT-B-32 | ViT-L-14 | Random | RKME | PAVE | Oracle |
|---|---|---|---|---|---|---|---|
| CIFAR10 | 0.100 | 0.899 | **0.956** | 0.780 | 0.511 | 0.890 | 0.979 |
| CIFAR100 | 0.008 | 0.663 | 0.777 | 0.527 | 0.226 | **0.885** | 0.885 |
| Cars | 0.006 | 0.604 | **0.784** | 0.492 | 0.462 | 0.680 | 0.680 |
| DTD | 0.022 | 0.440 | 0.556 | 0.406 | 0.293 | **0.698** | 0.698 |
| EuroSAT | 0.161 | 0.541 | 0.536 | 0.437 | 0.112 | **0.991** | 0.991 |
| Food101 | 0.008 | 0.828 | **0.924** | 0.699 | 0.455 | 0.886 | 0.886 |
| GTSRB | 0.030 | 0.330 | 0.511 | 0.351 | 0.295 | **0.987** | 0.987 |
| MNIST | 0.117 | 0.503 | 0.764 | 0.551 | 0.811 | **0.997** | 0.997 |
| RESISC45 | 0.019 | 0.611 | 0.713 | 0.524 | 0.338 | **0.934** | 0.934 |
| STL10 | 0.060 | 0.973 | **0.994** | 0.927 | 0.831 | 0.985 | 0.985 |
| SUN397 | 0.003 | 0.648 | 0.694 | 0.590 | 0.616 | **0.739** | 0.739 |
| SVHN | 0.096 | 0.329 | 0.584 | 0.398 | **0.970** | **0.970** | 0.970 |
| Avg.(↑) | 0.052 | 0.614 | 0.733 | 0.557 | 0.493 | **0.887** | 0.894 |
| Avg. rank(↓) | 5.917 | 3.250 | 1.917 | 4.000 | 4.417 | **1.417** | - |
| PAVE (win/tie/loss) | 12/0/0 | 11/0/1 | 8/0/4 | 12/0/0 | 11/1/0 | - | 0/11/1 |

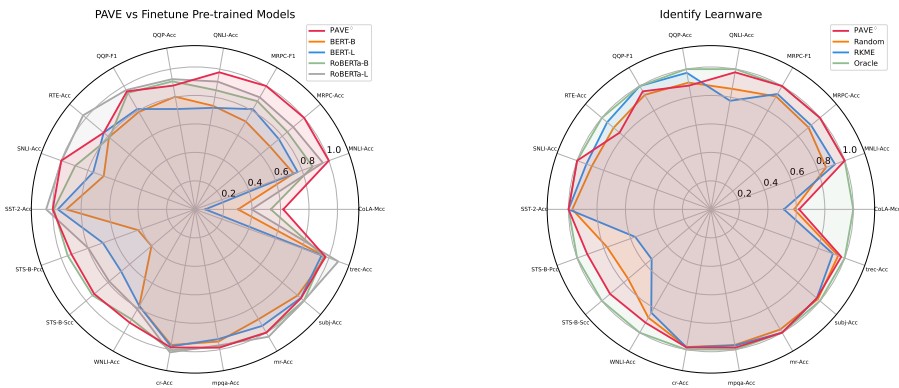

Figure A9: **Performances on classic NLP datasets beyond the original functionality.** And to achieve better visualization, all performance metrics are normalized relative to the Oracle.

## D.2 BEYOND THE ORIGINAL FUNCTIONALITY

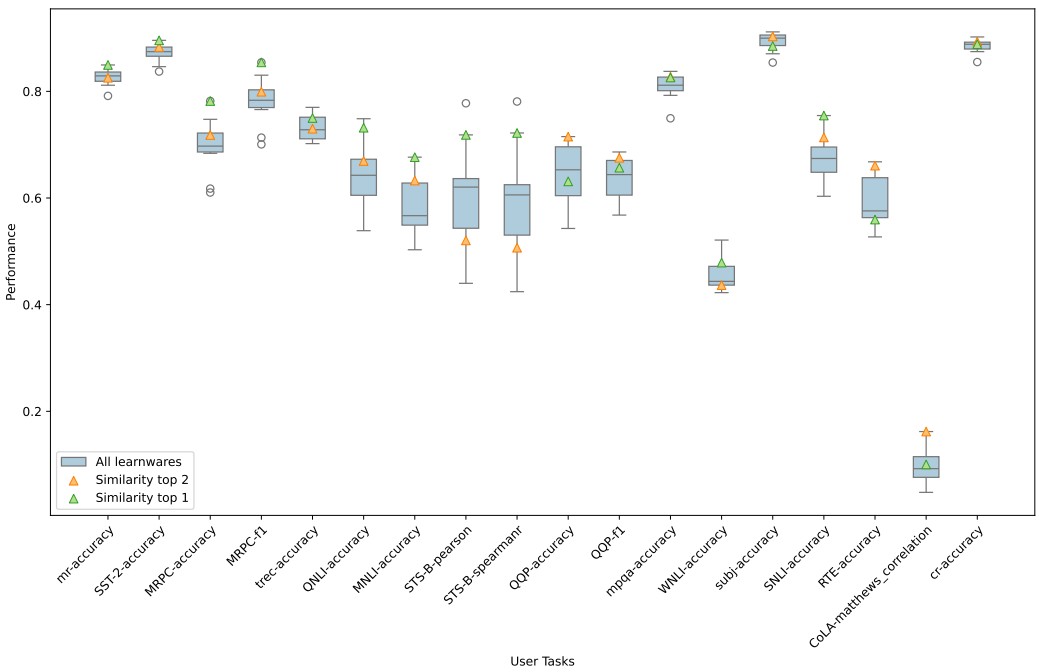

Figure A10: Visualization of performance on classic NLP datasets beyond the original functionality.

Table A7: Performances on computer vision datasets beyond the original functionality.

| User Task | Finetune Pre-trained Models | | | Identify Learnware | | | |
| --- | --- | --- | --- | --- | --- | --- | --- |
| Dataset | ResNet-152 | ViT-B-32 | ViT-L-14 | Random | RKME | PAVE$^\diamond$ | Oracle |
| CIFAR10 | 0.100 | 0.899 | **0.956** | 0.795 | 0.577 | 0.897 | 0.916 |
| CIFAR100 | 0.008 | 0.663 | **0.777** | 0.617 | 0.410 | 0.693 | 0.698 |
| Cars | 0.006 | 0.604 | **0.784** | 0.573 | 0.555 | 0.613 | 0.613 |
| DTD | 0.022 | 0.440 | **0.556** | 0.405 | 0.334 | 0.461 | 0.461 |
| EuroSAT | 0.161 | 0.541 | 0.536 | 0.605 | 0.182 | **0.782** | 0.782 |
| Food101 | 0.008 | 0.828 | **0.924** | 0.786 | 0.714 | 0.828 | 0.833 |
| GTSRB | 0.030 | 0.330 | **0.511** | 0.368 | 0.361 | 0.370 | 0.426 |
| MNIST | 0.117 | 0.503 | 0.764 | 0.627 | **0.864** | **0.864** | 0.864 |
| RESISC45 | 0.019 | 0.611 | **0.713** | 0.569 | 0.481 | 0.515 | 0.657 |
| STL10 | 0.060 | 0.973 | **0.994** | 0.932 | 0.853 | 0.931 | 0.975 |
| SUN397 | 0.003 | 0.648 | **0.694** | 0.617 | 0.628 | 0.638 | 0.638 |
| SVHN | 0.096 | 0.329 | **0.584** | 0.436 | 0.538 | 0.346 | 0.538 |
| Avg.($\uparrow$) | 0.052 | 0.614 | 0.733 | 0.611 | 0.541 | 0.662 | 0.700 |
| Avg. rank($\downarrow$) | 6.000 | 3.083 | 1.417 | 3.583 | 4.250 | 2.583 | - |
| PAVE$^\diamond$ (win/tie/loss) | 12/0/0 | 7/0/5 | 2/0/10 | 9/0/3 | 10/1/1 | - | 0/5/7 |

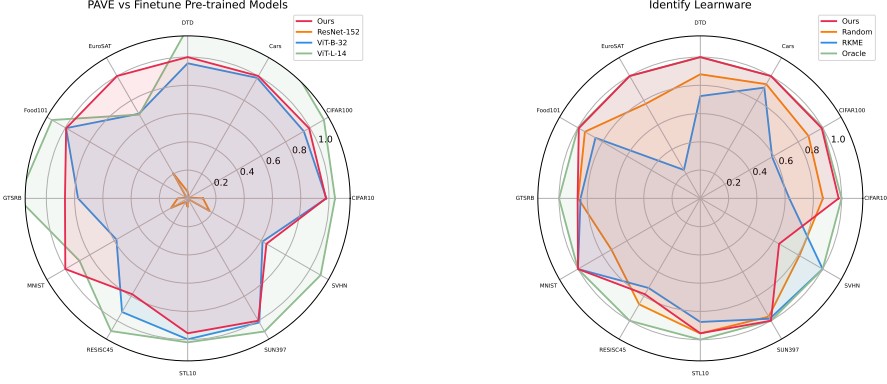

Figure A11: **Performances on computer vision datasets beyond the original functionality.** And to achieve better visualization, all performance metrics are normalized relative to the Oracle.

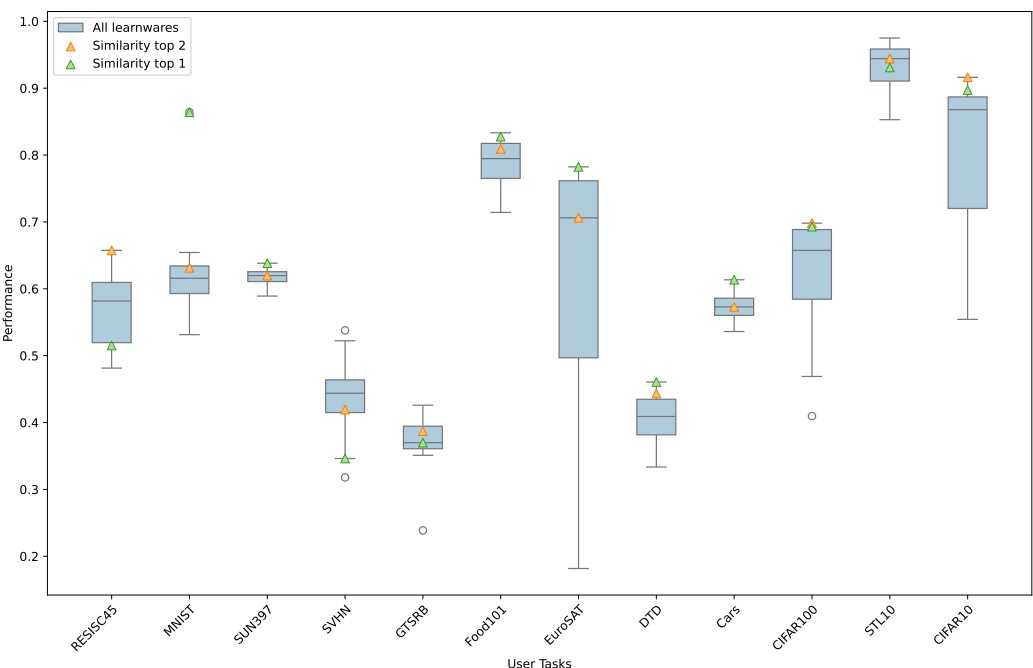

Figure A12: Visualization of performance on computer vision datasets beyond the original functionality.

## D.3 ABLATION STUDY

We provide supplementary results for the ablation study, including performance tables where "PAVE♣" represents the performance of task vectors derived from fitting the data distribution $p(y|\mathbf{x})$, serving as the ablation variant. Additionally, we present similarity matrices across tasks to show the effectiveness of parameter vector similarity approximation in the low-rank space. These materials offer insights into the contribution of key components and the robustness of our approach.

Table A8: Performances on classic NLP datasets with corrupted learnwares.

| User Task | | Finetune Pre-trained Models | | | | Identify Learnware | | | |
|---|---|---|---|---|---|---|---|---|---|
| Dataset | Metric | BERT-B | BERT-L | RoBERTa-B | RoBERTa-L | Random | PAVE♣ | PAVE | Oracle |
| CoLA | Mcc | 0.049 | 0.012 | 0.086 | 0.064 | -0.006 | 0.215 | **0.460** | 0.460 |
| MNLI | Acc | 0.498 | 0.519 | 0.594 | 0.646 | 0.505 | 0.611 | **0.731** | 0.731 |
| MRPC | Acc | 0.537 | 0.600 | 0.662 | 0.703 | 0.500 | 0.583 | **0.743** | 0.743 |
| MRPC | F1 | 0.609 | 0.694 | 0.749 | 0.776 | 0.604 | 0.675 | **0.811** | 0.827 |
| QNLI | Acc | 0.552 | 0.542 | 0.636 | 0.682 | 0.533 | 0.662 | **0.817** | 0.817 |
| QQP | Acc | 0.574 | 0.512 | 0.653 | 0.664 | 0.472 | 0.587 | **0.771** | 0.771 |
| QQP | F1 | 0.542 | 0.558 | 0.648 | 0.664 | 0.543 | 0.627 | **0.737** | 0.737 |
| RTE | Acc | 0.527 | 0.560 | 0.523 | 0.686 | 0.558 | 0.610 | **0.690** | 0.690 |
| SNLI | Acc | 0.515 | 0.575 | 0.676 | 0.753 | 0.502 | 0.635 | **0.785** | 0.785 |
| SST-2 | Acc | 0.811 | 0.864 | 0.889 | **0.937** | 0.789 | 0.846 | 0.908 | 0.917 |
| STS-B | Pcc | 0.332 | 0.536 | 0.740 | 0.620 | 0.230 | 0.501 | **0.859** | 0.859 |
| STS-B | Scc | 0.312 | 0.529 | 0.734 | 0.615 | 0.274 | 0.527 | **0.860** | 0.860 |
| WNLI | Acc | 0.408 | 0.408 | **0.465** | 0.423 | 0.420 | 0.338 | 0.268 | 0.493 |
| cr | Acc | 0.873 | 0.881 | 0.906 | 0.920 | 0.804 | 0.862 | **0.921** | 0.921 |
| mpqa | Acc | 0.789 | 0.775 | 0.822 | 0.812 | 0.706 | 0.790 | **0.869** | 0.869 |
| mr | Acc | 0.757 | 0.802 | 0.849 | **0.877** | 0.745 | 0.795 | 0.856 | 0.872 |
| subj | Acc | 0.858 | 0.883 | 0.910 | 0.909 | 0.574 | 0.736 | **0.940** | 0.940 |
| trec | Acc | 0.752 | 0.728 | 0.740 | 0.824 | 0.275 | 0.599 | **0.948** | 0.948 |
| Avg.(↑) | | 0.572 | 0.610 | 0.682 | 0.699 | 0.502 | 0.622 | **0.776** | 0.791 |
| Avg. rank(↓) | | 5.500 | 4.667 | 3.056 | 2.222 | 6.556 | 4.500 | **1.444** | - |
| PAVE (win/tie/loss) | | 17/0/1 | 17/0/1 | 17/0/1 | 15/0/3 | 17/0/1 | 17/0/1 | - | 0/14/4 |
| PAVE♣ (win/tie/loss) | | 14/0/4 | 8/0/10 | 4/0/14 | 1/0/17 | 17/0/1 | - | 1/0/17 | 0/0/18 |

Table A9: In mode (a), parameter similarity is computed exactly using the full parameter vectors obtained through complete fine-tuning. In mode (b), similarity is approximated in the low-rank space after expanding the parameter vectors to their full size as $(\mathbf{BA})_{m \times n}$. In mode (c), similarity is approximated both in the low-rank space and computed using only the matrix $\mathbf{B}$, which significantly reduces storage and computational costs—this is the method proposed in our approach.

| | Mode | Classic NLP Datasets | | Computer Vision Datasets | | Medical LLM Benchmarks | |
|---|---|---|---|---|---|---|---|
| | | Abs. | Ratio | Abs. | Ratio | Abs. | Ratio |
| Trainable Parameters | (a) | 120.61M | - | 147.08M | - | 472.55M | - |
| | (b) | 0.42M | 0.3% | 1.41M | 1.0% | 0.42M | 0.1% |
| | (c) | 0.42M | 0.3% | 1.41M | 1.0% | 0.42M | 0.1% |
| Computing Similarity | (a) | 120.61M | - | 147.08M | - | 472.55M | - |
| | (b) | 120.61M | 100.0% | 147.08M | 100.0% | 472.55M | 100.0% |
| | (c) | 0.42M | 0.3% | 1.41M | 1.0% | 0.42M | 0.1% |
| Storage Cost | (a) | 120.61M | - | 147.08M | - | 472.55M | - |
| | (b) | 0.42M | 0.3% | 1.41M | 1.0% | 0.42M | 0.1% |
| | (c) | 0.42M | 0.3% | 1.41M | 1.0% | 0.42M | 0.1% |
| Performance | (a) | 0.7807 | - | 0.8868 | - | 71.47 | - |
| | (b) | 0.7762 | 99.4% | 0.8868 | 100.0% | 72.78 | 101.8% |
| | (c) | 0.7762 | 99.4% | 0.8868 | 100.0% | 72.78 | 101.8% |

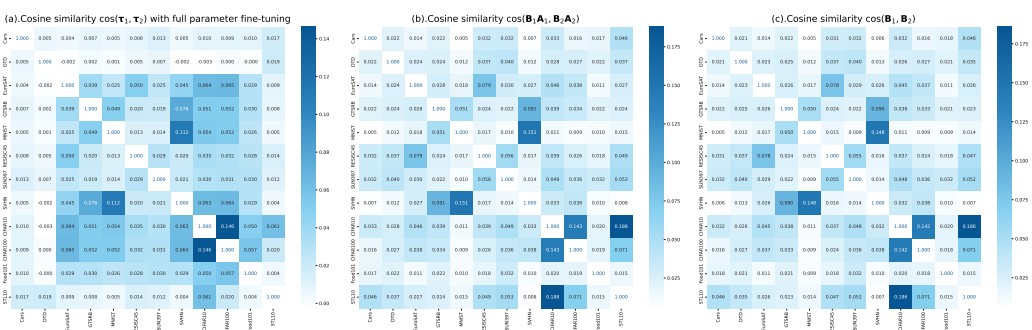

Figure A13: Consistency of the cosine similarity between parameter vectors on computer vision datasets, with the diagonal colors omitted to enhance visual contrast.

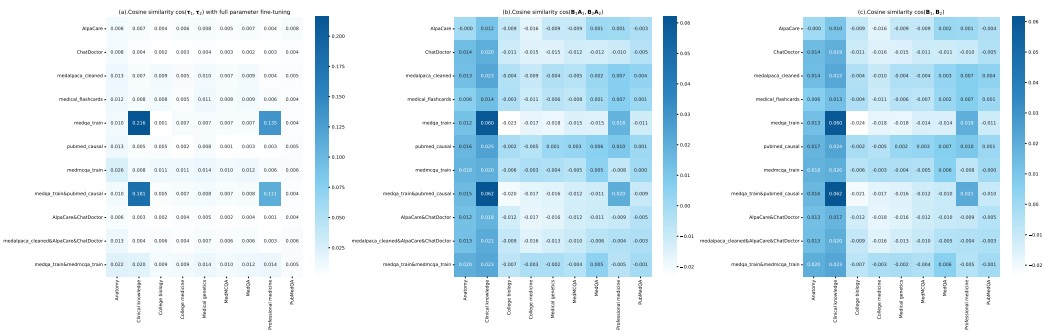

Figure A14: Consistency of the cosine similarity between model vectors on medical LLM benchmarks.

# E   ANALYSIS

## E.1   HIERARCHICAL LINEAR REGRESSION

To validate the effectiveness of our proposed method, we conduct a quantitative analysis to examine the relationship between similarity and performance. Specifically, we employ hierarchical linear regression to assess whether higher task vector similarity correlates with improved model performance on user tasks. Additionally, we use the superscripts "$\diamond$" and "$\clubsuit$" in the tables to denote different experimental settings, introduced in Section 4.2 and Section 4.3, respectively. These symbols help distinguish the specific configurations under which our method were evaluated. Note that although "PAVE$\clubsuit$" denotes the ablation variant in Section 4.3, it here represents the standard PAVE similarity in scenarios involving corrupted learnwares (i.e., those associated with low-quality models)

We conduct this analysis on both natural language processing (NLP) and computer vision (CV) datasets, treating the user task as a grouping variable (Woltman et al., 2012). The results, presented in Table A10 and Table A11 for NLP and CV, respectively, show a statistically significant positive correlation between task vector similarity and performance metrics. This indicates that models with higher similarity to the given task tend to achieve better results.

Table A10: Hierarchical linear regression on natural language processing datasets.

| | Coef. | Std Err. | z | P>|z| | CI Lower | CI Upper |
|---|---|---|---|---|---|---|
| Intercept | 0.508 | 0.053 | 9.557 | 0.000**** | 0.404 | 0.612 |
| Similarity | 0.788 | 0.063 | 12.482 | 0.000**** | 0.664 | 0.912 |
| Intercept$^{\diamond}$ | 0.667 | 0.053 | 12.644 | 0.000**** | 0.564 | 0.771 |
| Similarity$^{\diamond}$ | 0.122 | 0.044 | 2.785 | 0.005*** | 0.036 | 0.208 |
| Intercept$^{\clubsuit}$ | 0.499 | 0.057 | 8.688 | 0.000**** | 0.386 | 0.611 |
| Similarity$^{\clubsuit}$ | 0.708 | 0.047 | 15.050 | 0.000**** | 0.616 | 0.801 |

In the regression Table A10 and Table A11, the **P>|z|** column reports p-values, indicating the likelihood of the observed relationship under the null hypothesis. Lower values provide stronger evidence against the null, with common significance thresholds of $0.05$ and $0.01$[1]. In our analysis, significant p-values further validate the observed correlation.

Table A11: Hierarchical linear regression on computer vision datasets.

| | Coef. | Std Err. | z | P>|z| | CI Lower | CI Upper |
|---|---|---|---|---|---|---|
| Intercept | 0.497 | 0.051 | 9.703 | 0.000**** | 0.396 | 0.597 |
| Similarity | 2.988 | 0.282 | 10.612 | 0.000**** | 2.436 | 3.539 |
| Intercept$^{\diamond}$ | 0.592 | 0.049 | 12.138 | 0.000**** | 0.496 | 0.687 |
| Similarity$^{\diamond}$ | 1.738 | 0.572 | 3.040 | 0.002*** | 0.617 | 2.858 |
| Intercept$^{\clubsuit}$ | 0.557 | 0.055 | 10.089 | 0.000**** | 0.449 | 0.665 |
| Similarity$^{\clubsuit}$ | 2.248 | 0.227 | 9.911 | 0.000**** | 1.804 | 2.693 |

These findings provide strong empirical evidence for our approach, demonstrating that task similarity is a valuable criterion for selecting the most suitable learnware for a given user task. Furthermore, the consistency of results across different domains reinforces the generalizability of our method.

## E.2   EXACT MULTINOMIAL TEST

We employ the Exact Multinomial Test (Read and Cressie, 2012) to rigorously evaluate our proposed method against a null hypothesis of random selection. The Exact Multinomial Test is a non-parametric

---

[1]Significance levels: p $\leq$ 0.1 (*), p $\leq$ 0.05 (**), p $\leq$ 0.01 (***), p $\leq$ 0.001 (****).

statistical procedure used to determine whether the observed frequencies of categorical outcomes differ significantly from the frequencies expected under a given null hypothesis. In our context, each user task represents an independent trial where the outcome is defined by whether the learnware identified by our method is the optimal one for a given user task among those candidate learnwares.

Under the null hypothesis $\mathcal{H}_0$, we assume that each learnware has an equal probability of being identified for a given user task, which corresponds to the baseline method Random in our experiments. Therefore, we formally present the distribution $\pi_i$, which represents the number of top $k$ learnwares in representative performance on the user task among the top $k$ learnwares identified based on similarity from a candidate set of size $m$. The distribution $\pi_i$ is given as

$$\pi_i = \binom{m}{i} \left(\frac{k}{m}\right)^i \left(1 - \frac{k}{m}\right)^{k-i}, \quad \text{for } 0 \leq i \leq k,$$

where $i$ is the number of user tasks where our method is tied with Oracle in the evaluation results. And the variable $\mathbf{x}$ is the observation of the experiment, where $\mathbf{x}_i$ is the number of times the identification result with probability $\pi_i$ occurs. Then, we have

$$\Pr\left(\mathbf{x}\right)_0 = |\mathbf{x}|! \prod_{i=1}^{k} \frac{\pi_i^{x_i}}{x_i!}.$$

By calculating the exact probability of obtaining the observed evaluation results (or **a more extreme outcome**), the p-value is given as

$$p[\text{sig}] = \sum_{\mathbf{y}\,:\,\Pr(\mathbf{y})_0 \leq \Pr(\mathbf{x})_0} \Pr\left(\mathbf{y}\right)_0.$$

The Exact Multinomial Test allows us to determine if the performance of our method is statistically significant. This test is particularly useful in scenarios with discrete outcomes and limited sample sizes, ensuring robust inference without relying on large-sample approximations.

Table A12: The statistical significance of our method: p-values.

|  | Natural language processing Datasets | Computer Vision Datasets |
|---|---|---|
| PAVE | $0.00000^{****}$ | $0.00000^{****}$ |
| PAVE$^{\diamond}$ | $0.00081^{****}$ | $0.00193^{***}$ |
| PAVE$^{\clubsuit}$ | $0.00000^{****}$ | $0.00000^{****}$ |

Our analysis revealed that the frequency with which our method identifies the most suitable model is significantly higher than would be expected by chance, where the $k$ is set to $1$. Specifically, in both natural language processing and computer vision tasks, the p-values obtained from the Exact Multinomial Test indicate that the improvements are statistically significant, presented in Table A12. Note that statistical significance is commonly inferred when the p-value is less than $0.05$ or $0.01$[2]. These findings provide strong evidence that our method is effective in selecting superior models for user tasks, thereby validating its practical utility across different domains.

---

[2]Significance levels: $p \leq 0.1$ (*), $p \leq 0.05$ (**), $p \leq 0.01$ (***), $p \leq 0.001$ (****).

# F    DISCUSSION ON RKME

In this section, we first provide a brief overview of the Reduced Kernel Mean Embedding (RKME) specification. Then we highlight its similarities and differences with the Parameter Vector (PAVE) specification, emphasizing the unique strengths of our approach for deep learning models.

## F.1    REDUCED KERNEL MEAN EMBEDDING (RKME)

Specification is the central part of the learnware, capturing the model ability. We briefly introduce the Reduced Kernel Mean Embedding (RKME) specification (Zhou and Tan, 2024), which sketches the data distribution in the sample space $\mathcal{X} \times \mathcal{Y}$ of inputs and outputs with kernel methods.

We start by introducing the Kernel Mean Embedding (KME) (Schölkopf and Smola, 2002), which offers a novel representation for distributions. KME transforms a distribution into a reproducing kernel Hilbert space (RKHS). Given a distribution $\mathcal{D}$ defined over a space $\mathcal{X}$, the KME is defined as:

$$\mu_k(\mathcal{D}) \triangleq \int_{\mathcal{X}} k(\mathbf{x}, \cdot) d\mathcal{D}(\mathbf{x}), \tag{48}$$

where $k : \mathcal{X} \times \mathcal{X} \to \mathbb{R}$ is a symmetric and positive definite kernel function, and its associated RKHS is $\mathcal{H}$. For a data set $\{\mathbf{x}_i\}_{i=1}^m$ sampled from $\mathcal{D}$, the empirical estimate of KME is given by:

$$\hat{\mu}_k(\mathcal{D}) \triangleq \frac{1}{m} \sum_{i=1}^m k(\mathbf{x}_i, \cdot). \tag{49}$$

KME is considered as a potential specification due to several favorable properties. Accessing the raw data, however, compromises the necessary privacy concerns of the specification. Based on KME, the RKME specification is proposed to use **a synthetic dataset** of minor weighted samples $\{(\beta_j, t_j)\}_{j=1}^n, n \ll m$ to approximate the empirical KME of the original dataset with model pseudo-outputs $\{q_i\}_{i=1}^m = \{(\mathbf{x}_i, \hat{y}_i)\}_{i=1}^m$, where $\hat{y}_i = h(\mathbf{x}_i)$ is the model prediction. The synthetic dataset is generated through minimizing the Maximum Mean Discrepancy (MMD) (Gretton et al., 2012):

$$\min_{\beta, t} \left\| \frac{1}{m} \sum_{i=1}^m k(q_i, \cdot) - \sum_{j=1}^n \beta_j k(t_j, \cdot) \right\|_{\mathcal{H}}^2, \tag{50}$$

with the non-negative coefficients $\{\beta_j\}_{j=1}^n$. The RKME $\Phi(\cdot) = \sum_{j=1}^n \beta_j k(t_j, \cdot) \in \mathcal{H}$ acts as the specification, and the RKHS $\mathcal{H}$ is referred to as the specification space. This specification effectively captures the major information of the distribution $\mathcal{D}$ without exposing raw data.

## F.2    COMPARISON OF RKME AND PAVE

The RKME is initially designed for models on tabular data, whereas our method focuses on deep learning models. To protect data privacy, RKME generates a synthetic dataset to represent the data distribution rather than the original data via minimizing Maximum Mean Discrepancy (MMD) (Wu et al., 2023). However, for high-dimensional, unstructured data such as texts and images, generating meaningful synthetic data by minimizing MMD is still challenging and computationally expensive. In contrast, our method identify learnwares via the parameter vector similarity and has certain advantages in capturing the capabilities of deep learning models. Similarly, we compressed the parameter vector in a low-rank space, which intuitively prevents the recovery of the private original data. From a high-level perspective, both methods generate a model's representation—a specification—in the representation space induced by a helper function, $f$. In RKME, $f$ is a kernel function, and the specification is a reduced set representation in the RKHS spanned by $f$. In contrast, our proposed PAVE employs a pre-trained model as $f$, yielding a vector representation in the function space spanned by $f$. Below is a comprehensive breakdown of the differences.

**Application Scenarios.** RKME is primarily designed for tabular data, where kernel-based methods can effectively capture the underlying distribution. Its reliance on kernel mean embeddings makes it well-suited for structured datasets where feature relationships can be explicitly modeled. In contrast, our method is tailored for deep learning tasks, where high-dimensional features and complex

relationships between inputs and outputs are the norm. By leveraging pre-trained models and parameter vectors, our approach naturally adapts to a wide range of deep learning applications, making it more versatile in real-world deep learning tasks.

**Representation.** The two methods differ significantly in how they store and represent information. RKME implicitly encodes data distributions within a reproducing kernel Hilbert space (RKHS), approximating the empirical kernel mean embedding through synthetic data. This approach ensures privacy while retaining key distributional properties. Our method, on the other hand, directly represents model specifications in $\mathcal{H}$ using parameter vectors derived from pre-trained models. This direct approximation enables more efficient encoding of model capability and task information while circumventing the challenges of constructing high-quality synthetic data by minimizing MMD.

**Privacy.** Both methods prioritize data privacy but achieve it through different mechanisms. RKME protects raw data by generating a synthetic dataset, ensuring that the original data remains inaccessible while preserving statistical properties through maximum mean discrepancy minimization. Our method, instead, compresses information via low-rank approximations, making the reconstruction of the original dataset impossible. This not only safeguards privacy but also reduces storage and computational costs, making our approach more efficient for large-scale applications.

**Generalization.** RKME's reliance on symmetric kernel functions introduces limitations in capturing task semantics and model quality. When comparing tasks with different output spaces—such as binary classification versus multi-class classification—it becomes difficult to define a reasonable kernel function that respects symmetry while preserving meaningful task information. The problem is further exacerbated when comparing model outputs, as user tasks involve raw data while model predictions represent probability distributions, leading to inconsistencies in how similarity is measured. Our method overcomes these issues by constructing parameter vectors with the task's customized loss function, which provide a general way to generate specification across diverse task type and semantics. By measuring the similarity of parameter vectors, the compatibility between any model and task can be evaluated, regardless of divergences in task types and semantics.

**Expressiveness and efficiency.** RKME struggles to effectively capture sample similarities in text and image datasets due to the limited expressiveness of traditional kernel methods. While neural tangent kernels (NTKs) could theoretically enhance synthetic data generation, they come at a high computational cost. In contrast, our method leverages the structural advantages of pre-trained models, inherently benefiting from NTKs without added complexity. By operating directly in the pre-trained model's parameter space, our approach eliminates the inefficiencies of constructing synthetic datasets to approximate kernel mean embeddings. This enables a more expressive and scalable solution for deep learning tasks such as natural language processing and computer vision.

In summary, while RKME offers a privacy-preserving, kernel-based specification for tabular data, our method is better suited for deep learning models. It provides a more effective and universal representation of task semantics and model quality while maintaining greater computational efficiency.

## G  DISCUSSION ON MODEL DESCRIPTION

In the current landscape of LLMs, agents, and world models, model selection and coordination primarily rely on **semantic-level** model descriptions such as Model Cards (Mitchell et al., 2019) or the Model Context Protocol (MCP) (Anthropic, 2024). For instance, HuggingGPT (Shen et al., 2023) leverages LLMs as controllers to parse user requests and select expert models from Hugging Face based on their textual task descriptions. Extending this paradigm to tool use, Gorilla (Patil et al., 2024) and ToolLLM (Qin et al., 2024) align user instructions with API documentation to facilitate tool selection and invocation via retrieval-augmented generation or instruction tuning. While such semantic descriptions support high-level coordination through natural language or structured metadata, they remain inherently extrinsic and often fail to faithfully reflect actual model capabilities, leading to a "vocabulary gap" between textual specifications and empirical behaviors. To bridge this gap, PAVE characterizes models at the functional level as conditional distributions, providing an intrinsic representation that bypasses the ambiguity of textual metadata. This formulation enables mathematically grounded identification of helpful models across diverse manifestations, offering a level of precision and reliability that semantic descriptions fundamentally lack.

## H    LLM Usage Disclosure

Large Language Models were used solely to aid in polishing and editing the writing of this manuscript. Their role was limited to language refinement, such as improving clarity and grammar. No LLMs were employed in idea generation, experimental design, data collection, analysis, or code development. All scientific contributions are the original work of the authors.

