# OpenReview forum: "A Study on PAVE Specification for Learnware"
_ICLR.cc/2026/Conference — ICLR 2026 Poster_

### Official Review · Reviewer_iVnw · 2025-10-31

**Soundness:** 3
**Presentation:** 3
**Contribution:** 3
**Rating:** 8
**Confidence:** 2

**Summary:**

This paper introduces the Parameter Vector (PAVE) specification to identify and select high-quality learnwares for reuse in solving new user ML tasks. The PAVE approach formalizes the use of parameter vector changes (from fine-tuning a shared pre-trained model) as a representation of both a model’s capability and the requirements of the user’s task. Theoretically, the work connects PAVE to kernel mean embedding approaches via the neural tangent kernel regime, derives error bounds for low-rank approximations of the parameter vectors, and demonstrates through extensive empirical studies that PAVE achieves superior learnware selection, often outperforming conventional fine-tuning and previous learnware identification methods.

**Strengths:**

- The paper tackles the practical challenge of efficiently reusing high-performing models by providing a specification (PAVE) that allows users to identify suitable models without direct per-model evaluation.
- The mathematical exposition is detailed and generally clear.
- The proposal for low-rank approximation of parameter vectors significantly reduces memory and compute, a nontrivial contribution given the scale of modern check-pointed models.
- The experimental validation is exceptionally thorough and is a major strength of the paper.

**Weaknesses:**

Generally, I did not identify any major limitations or weaknesses in its core contributions. This is a high-quality paper that makes a practical contribution to the field of model reuse and the Learnware paradigm. The proposed PAVE specification is novel, well-motivated, and supported by both theoretical analysis and a comprehensive set of experiments.

**Questions:**

The PAVE specification, as described in L51-53, appears to rely on a shared pre-trained model and architecture to ensure the comparability of parameter vectors. Could the authors clarify if this is a necessary constraint? Furthermore, could the authors provide discussion on the potential for generalizing PAVE to a more heterogeneous setting, where learnwares in a repository might originate from different base models or possess distinct architectures?

---

> ### Author Response · Authors · 2025-11-24
>
> We appreciate R3's positive assessment and the recognition of our work as "high-quality" with "exceptionally thorough" experiments.
>
> Regarding your clarification request on the "shared pre-trained model" and the "heterogeneous setting": our method *does* support this setting. The learnware models in the repository can originate from different base models and possess distinct architectures (e.g., BERT-large, RoBERTa, LLaMa, etc.).
>
> The 'shared model' constraint applies *only* to the separate, public model (e.g., **ReBERTa**) used to generate the specification vectors. This is not a limitation but the core feature that makes PAVE **generalizable** across different learnwares. We use this single **ReBERTa** model in two ways:
>
> 1. **To create learnware specifications:** We fine-tune it to mimic the predictions of each heterogeneous learnware.
> 2. **To create the task** **vector****:** We fine-tune the *exact same model* on the user's few-shot data.
>
> Because both the learnware specifications and the task vector are parameter vectors from this *same* **ReBERTa** model, they live in the same vector space and can be meaningfully compared.
>
> This separation between the learnware's architecture and the specification's architecture is a key strength. We will revise our Methodology (Section 3) to make this distinction more explicit.
>
> Thank you again for your strong support.

---

> > ### Comment · Reviewer_iVnw · 2025-11-25
> >
> > Thank you for your response. I would like to keep my original score, given the practical challenges that the authors are trying to address.

---

### Official Review · Reviewer_bGit · 2025-11-01

**Soundness:** 3
**Presentation:** 3
**Contribution:** 3
**Rating:** 6
**Confidence:** 2

**Summary:**

The paper proposes a new way, called PAVE (Parameter Vector), to describe and identify reusable models in the Learnware framework. Instead of using reduced data samples to represent a model’s capability (as in prior work like RKME), PAVE uses how a model’s parameters change during fine-tuning. This parameter-based “signature” helps match models (learnwares) with user tasks more efficiently and works even when data are high-dimensional (e.g., text or images). The authors also derive a theoretical link between PAVE and RKME through the neural tangent kernel (NTK) and propose a low-rank approximation to make computation feasible. Experiments on NLP and CV tasks show that PAVE can identify suitable learnwares better than baselines and sometimes outperform fine-tuned pre-trained models

**Strengths:**

1. Using parameter changes instead of data samples to represent a model’s capability is original and intuitive, especially for privacy-sensitive or unstructured data.
2. The connection between PAVE and RKME under the NTK assumption provides a sound theoretical explanation for why the approach should work.
3. The low-rank approximation (similar to LoRA) reduces the storage and computational cost while maintaining similarity accuracy
4. The experiments cover multiple domains (NLP, CV, medical LLMs) and show clear and consistent improvements over RKME and fine-tuning baselines.

**Weaknesses:**

1. While the paper introduces PAVE as a general solution, it’s not entirely clear how broadly this parameter-vector representation generalizes beyond fine-tuned models based on shared backbones.
2. The method assumes all learnwares are fine-tuned from a common pre-trained model. It’s unclear how it performs when models come from different architectures or pre-training distributions.
3. The experiments mainly compare to RKME and basic fine-tuning baselines. It would strengthen the evaluation to include more data-centric reuse or transferability estimation methods, such as LEEP, LogME, or task2vec, to contextualize PAVE’s advantage.
4. The paper doesn’t discuss how PAVE specifications would be stored, shared, or updated in a real learnware repository, or how they interact with privacy and security constraints.

**Questions:**

Please refer to the weakness section

---

> ### Author Response · Authors · 2025-11-24
>
> We thank R2 for the positive and very constructive feedback.
>
> **1.Shared Model (Main Weakness):** We must clarify a central misunderstanding regarding the "shared model." The concern that "all learnwares are fine-tuned from a common pre-trained model" **is inaccurate**.
>
> Our method **explicitly supports a heterogeneous** **repository** where learnwares can have different architectures (e.g., BERT-large, RoBERTa, LLaMa, etc.).
>
> The misunderstanding likely arises from our use of a **"shared pretrained model"** (e.g., ReBERTa in our experiments). This model is **not** the learnware itself. It is a *separate, public model* used **only** to generate the specification vectors (PAVEs). The workflow is:
>
> - **Step 1. For Learnwares:** We fine-tune this **shared pretrained model** (ReBERTa) to mimic the predictions of *each* heterogeneous learnware (e.g., a BERT-large model, a RoBERTa model, etc.).
> - **Step 2. For User Tasks:** We fine-tune the *exact same* **shared pretrained model** (ReBERTa) on the user's few-shot data.
>
> Because all model vectors (from step 1) and the task vector (from step 2) are parameter vectors from the same ReBERTa model, they live in the same vector space and can be meaningfully compared.
>
> This **separation between the** ***learnware's*** **architecture and the** ***specification's*** **architecture** is a key strength of our paradigm. We will emphatically revise our Methodology section (**Section 3**) to make this distinction crystal clear. We trust this resolves the primary concern.
>
> **2.Baselines (LEEP, LogME, Task2vec):** As noted in our Related Work (Transferability subsection), these methods are fundamentally different from our approach.
>
> - **LEEP** requires running the target dataset through the pretrained model (PTM) once to estimate transferability via a log-expectation–based proxy metric.
> - **LogME** similarly depends on a forward pass of the PTM on the target dataset to extract PTM-specific features and compute the marginalized likelihood.
> - **Task2vec** is even more computationally demanding: it requires training a large reference network (called the probe network), adapting it to the target data set, and computing the Fisher information matrix to obtain a task embedding.
>
> These baseline estimators share two fundamental drawbacks: 1). **High Computational Cost:** They all require expensive operations on the target data, ranging from full model inference and feature extraction (LEEP, LogME) to additional network training and matrix computation (Task2vec). 2). **Data Privacy Violation:** They necessitate **direct access to raw target data** to perform these computations, raising significant privacy concerns.
>
> **Our learnware-based method** directly solves both: 1). **Efficiency:** Our method uses specifications (PAVEs). Model matching is reduced to a simple, fast **cosine similarity** **calculation** between PAVE vectors, completely avoiding costly inference or training. 2). **Privacy:** The user's task PAVE is **generated locally** on the user's side. Raw data is never shared with the online system, preserving data privacy by design.
>
> **3.Storage & Privacy:** These are important practical points, and they are core to our design.
>
> - **Storage & Retrieval.** As we noted in our **Introduction** (as a core contribution) and detailed in **Section 3.3**, the low-rank approximation (similar to LoRA) makes PAVE specifications extremely small (~1MB). This compact vector format is ideal for storage and enables efficient, large-scale retrieval using **standard vector databases**.
> - **Privacy.** PAVE also offers a significant privacy advantage over the data-dependent baselines R2 suggested (e.g., LEEP, LogME, task2vec). Those methods require access to the user's (potentially sensitive) task data to compute a score. In contrast, with PAVE, **the user's data never leaves their side; they only need to submit the resulting low-rank parameter deltas** (the vector). This **low-rank approximation itself enhances privacy**, as it is a highly compressed, abstract representation of the task, inherently obscuring fine-grained details from the original data. We plan to conduct a deeper analysis of these formal privacy-preserving effects as part of our future work.
>
> We hope these clarifications and revisions fully address your concerns.

---

### Official Review · Reviewer_KAPE · 2025-11-03

**Soundness:** 3
**Presentation:** 3
**Contribution:** 3
**Rating:** 4
**Confidence:** 1

**Summary:**

The paper considers a setup that we have access to a large number of models. Each model consists of a model and a "specification" that describes its capabilities. The central problem is how to create a specification that allows a user to efficiently identify the most helpful model from a vast repository, without the costly process of evaluating every single model.

The main idea is simple: assume we have access to a trained model h. To create its specification, they take a shared, public pre-trained model (e.g., BERT, CLIP) and fine-tune it to mimic the predictions of their model h. The resulting change in the pre-trained model's parameters is saved as the model vector.  A user with a new task provides a small, few-shot dataset. To create their task specification, they fine-tune the exact same shared pre-trained model on their few-shot data to fit the true labels. This parameter change becomes the task vector. Finally the selection is based on measuring the similarity between task vector and the finetuning vector.

**Strengths:**

The PAVE method's primary strength is its effectiveness with high-dimensional, unstructured data like images and text, a scenario where prior specifications failed. The authors conducted extensive experiments across Natural Language Processing (NLP),

**Weaknesses:**

Reliance on a Shared Pre-trained Model: The entire system fundamentally relies on both the developers (creating learnwares) and the users (sketching their tasks) using the exact same shared pre-trained model as a common basis to generate the parameter vectors. I think the authors should study the robustness of their finding s to this assumptions.

**Questions:**

see above

---

> ### Author Response · Authors · 2025-11-24
>
> We thank R1 for the insightful review. Your question regarding the "reliance on the exact same shared model" and its **robustness** is an excellent and deep point.
>
> Regarding the **necessity** of a shared model, its purpose is to establish a "common basis." This ensures that the vector space for learnware specifications and task specifications is perfectly aligned, making their similarity scores meaningful.
>
> Regarding **robustness**—a core concern you raised—we emphasize that this aspect was thoroughly investigated in our submission. Our results confirm that the alignment is highly stable and the choice of the shared pre-trained model is not a fragile constraint. Specifically, we validated the method's effectiveness across **diverse task domains**, its stability under **unified hyperparameters**, and its robustness to **architectural discrepancies** between the shared model and learnwares. This is established by the findings reported in the paper:
>
> 1. **Robustness to Diverse Tasks:** Our experiments were conducted across several broad domains (e.g., NLP, CV). For each domain, we used a single **shared pretrained model** that proved highly effective for a **vast and diverse range of tasks** within that domain (shown in **Appendix C.1**). This *same* model effectively generated specifications for all the different tasks in our study, achieving strong results (as shown in **Section 4.2.1, 4.2.2, 4.3.1**).
> 2. **Stability Across Hyperparameters:** Critically, we used the *same set of finetuning hyperparameters* for the **shared pretrained model** across all different user tasks and learnware models (reported in **Table A3**). This demonstrates that our method is not sensitive to task-specific hyperparameter tuning, further proving its stability.
> 3. **Robustness to Architectural Differences:** We also show that the **shared pretrained model** does not need to share the *exact same architecture* as the learnwares. Our experiments demonstrate this across different model families. (We provide further details on the model architectures in Appendix C.3.) For instance, in **Section 4.2** and **Table 1, A5, A7, A9** , we successfully used a **RoBERTa**-based **shared pretrained model** to generate model vectors for various **BERT**-based learnwares. Separately, in our experiments with Large Language Models (**Table A4, A6**), we also successfully used a **Qwen-0.5B** model to generate specifications for **Llama**-based learnwares. The ability to correctly rank these architecturally distinct models demonstrates that our specification process is highly robust to architectural mismatches.

---

### Author Response · Authors · 2025-11-24

Dear Reviewers and Area Chair,

We sincerely thank you for your time and constructive feedback. We are highly encouraged by the positive assessment of our PAVE method, which reviewers found **"original and intuitive" (R2)** and backed by **"exceptionally thorough" experiments (R3)**.

We have carefully considered every point raised. Detailed, point-by-point responses to all comments and queries are provided in the dedicated sections that follow.

We appreciate your valuable input and look forward to your consideration.

---

### Meta-Review · Area_Chair_9fSB · 2026-01-08

**Summary:**

This paper addresses the challenge of efficiently identifying helpful pre-trained models (learnwares) without costly per-model evaluations. It proposes PAVE (Parameter Vector), a specification for learnware identification that utilizes the changes in pre-trained model parameters to inherently encode the model capability and task requirements. The reviewers praised the relevance of the problem, the theoretical grounding connecting PAVE to NTK (Neural Tangent Kernel), the usage of low-rank approximation, and the extensive experimental validation across NLP, CV, and medical domains. They also raised concerns on the reliance on a shared pre-trained model and the lack of comparison to a few methods in the literature (LEEP, LogME, task2vec).

**Reviewer Concerns:**

The authors provided detailed rebuttal to all comments, clarifying the usage of a shared pre-trained model (only to generate PAVEs) and the differences of their approach to those methods mentioned in the literature.

**Reviewer Scores:**

Reviewer iVnw has an initial score of 8. They responded to the rebuttal and decided to maintain their score. Reviewers bGit has an initial score of 6. They did not respond and would likely maintain or increase their score given the clarifications from the rebuttal. Reviewer KAPE has an initial score of 4, and might increase their score given the clarification in the usage of a shared pre-trained model. One thing to note is that all reviewers have relatively low confidence level (1-2).

---

### Decision · Program_Chairs · 2026-01-26

Accept (Poster)